

# Transcription levels and prognostic significance of the NFI family members in human cancers

Yuexian Li[1], Cheng Sun[1], Yonggang Tan[1], Lin Li[2], Heying Zhang[1], Yusi Liang[1], Juan Zeng[1] and Huawei Zou[1]

[1] The First Oncology Department, Shengjing Hospital affiliated with China Medical University, Shenyang, China
[2] The First Oncology Department, The Fourth Hospital affiliated with China Medical University, Shenyang, China

## ABSTRACT

**Background:** The nuclear factor I (NFI) is a family of transcription factors consisting of four distinct but closely related genes, NFIA, NFIB, NFIC and NFIX, which are important in the development of various tissues and organs in mammals. Recent study results have shown that NFI family may play a critical role in the progression of various human tumors and have been identified as key tumor suppressors and oncogenes for many cancers. However, the expression levels and distinctive prognostic values of the NFI family remain poorly explored in most cancers.
**Materials and Methods:** In the present study, the differences in mRNA expression of the NFI family in various cancers were investigated using the Oncomine and TCGA databases, and the mRNA expression, genetic alteration and DNA methylation of the NFI family members in various cancers were examined using cBioPortal for Cancer Genomics. In addition, the prognostic significance of the NFI family was assessed in multiple cancers using the Kaplan–Meier plotter (KM plotter) and SurvExpress databases.
**Results:** The mRNA expression levels in the NFI family were significantly downregulated in most cancers compared with normal tissues and DNA hypermethylation might downregulate the NFI family expression. Although NFIX expression was not downregulated in kidney, colorectal and prostate cancers. Furthermore, NFIB expression was upregulated in gastric cancer. Further survival analyses based on the KM plotter and SurvExpress databases showed dysregulations of the NFI genes were significantly correlated with survival outcomes in breast, lung, and head and neck cancers. Decreased expression levels of NFIA, NFIB and NFIC were associated with poor overall survival (OS) in head and neck cancer. Low mRNA expression of NFIA and NFIB was significantly associated with OS and first progression in lung adenocarcinoma, but not in lung squamous cell carcinoma. In addition, potential correlations between NFI family members and survival outcomes were also observed in liver, esophageal, kidney and cervical cancer.
**Conclusion:** The results from the present study indicated certain members of the NFI family could be promising therapeutic targets and novel prognostic biomarkers for human cancers.

Corresponding author
Huawei Zou, Zouhw999@163.com

## INTRODUCTION

Cancer is the leading cause of death and a major public health concern worldwide. Globally, 18.1 million new cancer cases and 9.6 million cancer deaths were reported in 2018 (*Bray et al., 2018*). In addition to the diagnostic techniques and treatments such as surgical resection, radiotherapy and new targeted chemotherapies have become more advanced. However, the efficacy of cancer treatments remains unsatisfactory. Thus, investigating the mechanisms of tumorigenesis and tumor progression is urgently needed, as well as identifying potential biomarkers for improved diagnosis, prognosis and treatment.

Nuclear factor I (NFI), or CCAAT box-binding transcription factor (CTF), was first identified as a single protein purified from human Hela cells and essential for the replication of adenovirus DNA in vitro (*Nagata et al., 1982*). In humans, the NFI family consists of four closely related transcription factors, NFIA, NFIB, NFIC and NFIX that can bind as either hetero- or homodimers to a duplex consensus sequence TTGGC(N5) GCCAA. These dimers have comparable affinity for DNA, stability, and specificity (*Kruse & Sippel, 1994*; *Leegwater, Van Driel & Van Der Vliet, 1985*). Over the past decades, the members of NFI family have been shown to regulate cell proliferation and differentiation during the development of multiple organ systems. Emerging evidence has gradually shown NFI expression in various cancers. In addition, results from other studies indicated that NFI genes are closely related to a number of tumor suppressor or oncogene processes and disease states. *Song et al. (2010)* demonstrated NFIA was highly expressed in astrocytomas and associated with better progression-free survival (PFS). NFIB is amplified and expressed in human small cell lung cancer (SCLC) and controls cell proliferation and apoptosis (*Dooley et al., 2011*). Moreover, lower NFIC expression levels were observed in breast cancer cells and exerted an inhibitory effect on the epithelial-mesenchymal transition (EMT), migration and invasion (*Lee, Lee & Park, 2015*). NFIX mRNA expression was downregulated in non-SCLC (NSCLC), and reduced NFIX expression was shown to independently predict poor prognosis in lung adenocarcinoma but not in squamous cell carcinoma (*Ge et al., 2018*). In summary, the results from previous studies indicate the NFI family members participate in multiple human cancers and may act as potential therapeutic targets or prognostic biomarkers in some cancers. However, a systematic analysis regarding the transcriptional expression and prognostic values in human cancers is lacking.

In the present study, the differences in mRNA expression of the NFI family members between tumors and normal tissues in multiple cancers were investigated using the Oncomine and TCGA databases. Furthermore, the mRNA expression, genetic alteration, and DNA methylation of the NFI family members in various cancers were examined using cBioPortal for Cancer Genomics. In addition, the prognostic significance of the NFI family was evaluated using the Kaplan–Meier Plotter (KM plotter) and SurvExpress databases.

## MATERIALS AND METHODS

### Oncomine database

Oncomine (http://www.oncomine.org), an online cancer microarray database and web-based data-mining platform, was used to analyze the individual mRNA levels of NFI transcription factors between cancers and respective normal tissues in multiple cancer types (*Rhodes et al., 2007*, *2004*). In this study, the thresholds were restricted as follows: *p*-value: 0.01; fold change: 2; gene rank: 10%; data type: mRNA; analysis type: cancer vs. normal analysis. Cancer type, sample size, fold change, *t*-test and *p*-value were obtained from studies that showed statistically significant differences.

### TCGA analysis using UCSC Xena browser

Integrin mRNA HiSeq expression data from the TCGA database involving breast, lung, melanoma, pancreatic and bladder cancers, as well as other cancers, were obtained from the UCSC Xena browser (https://xenabrowser.net) version: 2017-05-06. Student's *t*-test was performed to investigate differences in the mRNA expression levels between tumors and normal tissues. The boxplots were made using the GraphPad prism software.

### cBioPortal for Cancer Genomics database

The cBioPortal for Cancer Genomics (http://www.cbioportal.org/) is an open-access resource for the interactive exploration of multidimensional cancer genomics data sets. The genetic alterations in multiple cancers were examined using cBioPortal for Cancer Genomics (*Cerami et al., 2012*; *Gao et al., 2013*). The correlation between mRNA (RNA Seq V2 RSEM) and DNA methylation (HM450) in various cancers was calculated according to the cBioPortal's online instructions.

### KM plotter database

The KM plotter (http://kmplot.com/analysis/) database assesses the effects of 54,675 genes on survival in 18,674 cancer types. In this database, the survival data for breast, lung, bladder, head and neck, esophageal, and kidney cancers are available (*Nagy et al., 2018*). In the present study, the database was used to analyze the prognostic values of NFI genes in those cancers. For each gene symbol, the desired probe IDs were individually entered into the database to obtain KM plots. Patients were divided into high and low expression groups based on the median values of mRNA expression levels, and survival analyses were performed without follow-up restrictions. The number of cases, hazard ratios (HRs), 95% confidence intervals (CIs), and log rank *p*-values were extracted from the KM plotter webpage.

### SurvExpress database

The SurvExpress database was used to obtain survival data for prostate cancer, for which information was not available in the KM plotter database (*Aguirre-Gamboa et al., 2013*). The TCGA database was used for analysis because both the desirable probes and larger sample size were present (>200 patients). The hazard odds ratio with 95% CI having *p*-values ≤ 0.05 was considered statistically significant.

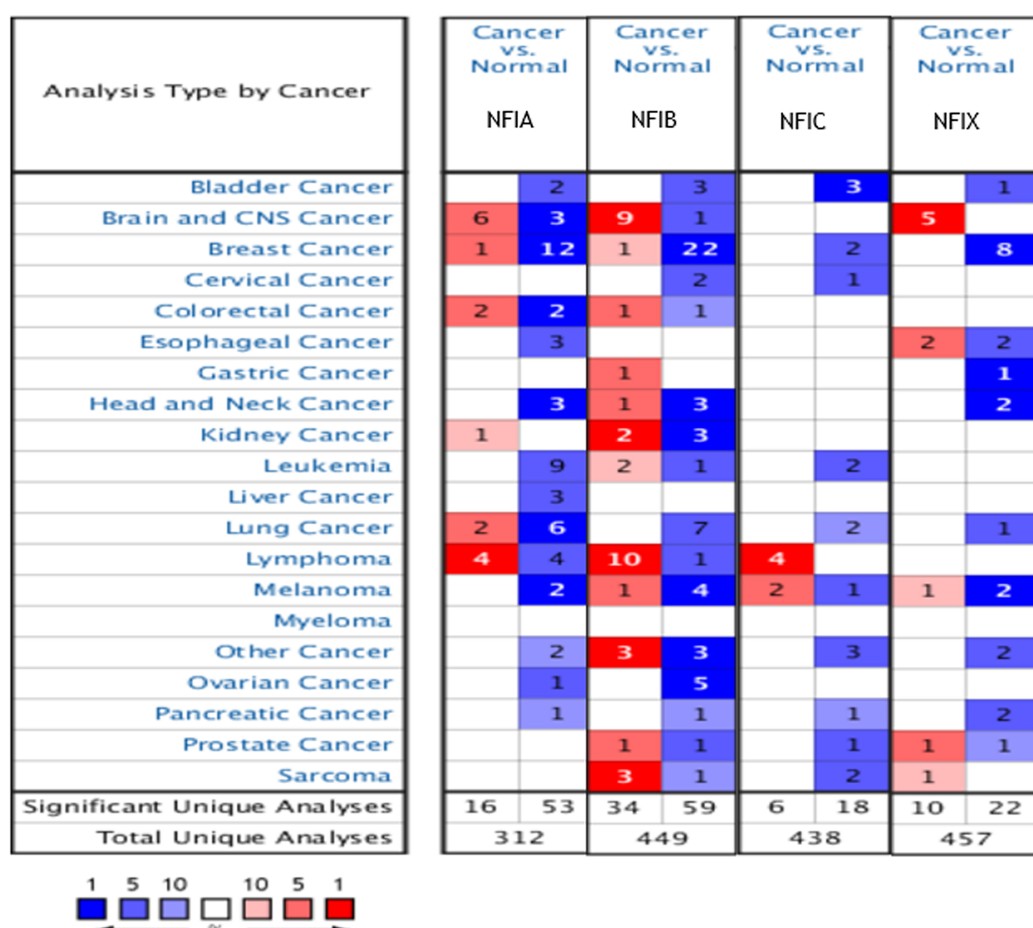

**Figure 1 The transcription levels of the NFI family members in different types of human cancers.** The figure is generated from ONCOMINE with extract thresholds (*p*-value: 0.01; fold change: 2 and gene rank: 10%). The cell number represents the dataset number that meets all of the thresholds with the color blue for underexpression and color red for overexpression. Cell color is determined by the best gene rank percentile for the analyses within the cell. NFI, Nuclear factor I; CNS, central nervous system.

## RESULTS

### The mRNA expression patterns of the NFI family members in human cancers

The Oncomine database was used to analyze the mRNA expression differences of four NFI genes between tumors and normal tissues in various cancers. As shown in Fig. 1, the database contained a total of 312, 449, 438 and 457 unique analyses for NFIA, NFIB, NFIC and NFIX, respectively. In 69 studies, a statistically significant difference for NFIA was observed. In 53 of the 69 studies, 14 types of cancers showed decreased NFIA mRNA expression level compared with normal tissues; however, in 16 studies, the opposite results were observed. Ninety-three unique analyses revealed the NFIB mRNA expression level varied with the type of tumor. Compared with normal tissues, NFIC mRNA expression level was reduced in tumors in 18 studies involving 10 types of cancers, however, an

increased level was observed in lymphomas and melanomas in only six studies. Regarding NFIX, 22 datasets revealed lower NFIX expression levels in 10 types of carcinomas with statistical significance, however, 10 analyses showed higher NFIX expression levels in brain and CNS cancer, esophageal cancer, melanoma, prostate cancer and sarcoma. Taken together, most of the analyses showed the NFI transcriptional expression levels were significantly reduced in tumors compared with normal tissues.

## Transcription levels and prognostic significance of the NFI family members in breast cancer

The mRNA expression levels of the NFI family members were first analyzed in breast cancer using the Oncomine database, which utilizes differential expression analyses by comparing most major types of cancer with respective normal tissues. In a total of 13 datasets, the differences in mRNA expression levels were compared between breast cancer and normal tissues. Analyses were available for NFIA, NFIB and NFIX in all 13 datasets, and for NFIC in 12 datasets. The NFIA mRNA level was found significantly downregulated in numerous databases including Kamoub (*Karnoub et al., 2007*), Richardson2 (*Richardson et al., 2006*) Curtis (*Curtis et al., 2012*), TCGA and Gluck (*Gluck et al., 2012*). However, the NFIA mRNA level was upregulated in invasive breast carcinoma in Finak's database (*Finak et al., 2008*). NFIB was significantly downregulated in 22 unique analyses across different breast cancer subtypes in 10 different databases including TCGA, Curtis (*Curtis et al., 2012*), Ma 4 (*Ma et al., 2009*), Zhao (*Zhao et al., 2004*), Turashvili (*Turashvili et al., 2007*), Sorlie (*Sorlie et al., 2001*), Richardson2 (*Richardson et al., 2006*), Sorlie2 (*Sorlie et al., 2003*), Perou and Gluck (*Gluck et al., 2012*; *Perou et al., 2000*). NFIC mRNA level was decreased in ductal breast carcinoma and lobular breast carcinoma in studies in which Richardson2 and Sorlie2 databases were utilized (*Richardson et al., 2006*; *Sorlie et al., 2003*). NFIX mRNA level was significantly reduced in multiple databases including Curtis (*Curtis et al., 2012*), Zhao (*Zhao et al., 2004*), Kamoub and Gluck (*Gluck et al., 2012*; *Karnoub et al., 2007*) for invasive ductal breast carcinoma, invasive ductal and invasive lobular breast carcinoma, invasive ductal breast carcinoma and invasive breast carcinoma, respectively. NFIX mRNA level was also significantly downregulated in ductal breast carcinoma compared with normal tissues in Sorlie (*Sorlie et al., 2001*), Perou and Sorlie2 databases (*Perou et al., 2000*; *Sorlie et al., 2003*). The statistically significant results are summarized in Table 1. Next, the mRNA HiSeq expression data from TCGA database was utilized to further determine the expression of the NFI family members in breast cancer. As shown in Fig. 2A, expression of all NFI family members was significantly downregulated in 1,104 cases of breast cancer compared with 114 normal samples. Next, the underlying mechanism of dysregulated expression of the NFI family was investigated using the cBioPortal online tool for breast invasive carcinoma (TCGA, Firehose Legacy). NFI genes were altered in 242 samples from 963 patients (25%) with breast invasive carcinoma. Specifically, genetic alteration of the NFI genes was analyzed and depicted as oncoprints representing mutation, amplification, deep deletion, mRNA high, mRNA low and multiple alterations (Figs. 2B and 2C). Survival analysis of the NFI genes with and without each gene alteration was

**Table 1 Datasets of the NFI family in breast cancer (ONCOMINE database).**

| Gene | Dataset | Normal (cases) | Tssumor (cases) | Fold change | t-Test | p-Value |
|------|---------|----------------|-----------------|-------------|--------|---------|
| NFIA | Karnoub | Breast (15) | Invasive ductal breast carcinoma (7) | −3.381 | −7.532 | 1.09E−06 |
| | Richardson 2 | Breast (7) | Ductal breast carcinoma (40) | −6.061 | −7.225 | 2.40E−09 |
| | Curtis | Breast (144) | Ductal breast carcinoma in situ (10) | −2.207 | −9.411 | 8.54E−07 |
| | | | Invasive ductal breast carcinoma (1,556) | −2.017 | −26.559 | 4.91E−69 |
| | | | Invasive breast carcinoma (21) | −2.292 | −8.448 | 9.48E−09 |
| | | | Medullary breast carcinoma (32) | −3.457 | −14.097 | 4.20E−17 |
| | | | Tubular breast carcinoma (67) | −2.039 | −15.725 | 3.61E−31 |
| | | | Mucinous breast carcinoma (46) | −2.227 | −14.215 | 4.25E−22 |
| | | | Breast carcinoma (14) | −2.137 | −6.042 | 1.51E−05 |
| | TCGA | Breast (61) | Invasive ductal breast carcinoma (389) | −2.277 | −11.49 | 7.00E−22 |
| | | | Mucinous breast carcinoma (4) | −2.851 | −10.414 | 1.96E−05 |
| | Gluck | Breast (4) | Invasive breast carcinoma (154) | −2.207 | −6.408 | 4.27E−04 |
| | Finak | Breast (6) | Invasive breast carcinoma (53) | 5.152 | 13.608 | 6.52E−18 |
| NFIB | TCGA | Breast (61) | Invasive ductal breast carcinoma (389) | −6.159 | −27.478 | 6.59E−80 |
| | | | Invasive breast carcinoma (76) | −4.54 | −13.706 | 5.59E−25 |
| | | | Invasive lobular breast carcinoma (36) | −4.361 | −9.378 | 2.28E−12 |
| | Curtis | Breast (144) | Invasive ductal breast carcinoma (1,556) | −3.689 | −37.562 | 7.71E−116 |
| | | | Invasive lobular breast carcinoma (148) | −2.675 | −16.623 | 1.54E−41 |
| | | | Invasive ductal and invasive lobular breast carcinoma (90) | −3.334 | −14.802 | 1.13E−28 |
| | | | Ductal breast carcinoma in situ (10) | −2.275 | −4.665 | 5.10E−04 |
| | | | Mucinous breast carcinoma (46) | −5.352 | −13.363 | 1.95E−18 |
| | | | Tubular breast carcinoma (67) | −3.096 | −13.05 | 4.36E−22 |
| | | | Breast phyllodes tumor (5) | −2.34 | −3.955 | 0.008 |
| | Ma 4 | Breast (14) | Invasive ductal breast carcinoma stroma (9) | −3.974 | −5.874 | 5.06E−06 |
| | | | Ductal breast carcinoma in situ epithelial (9) | −4.958 | −8.387 | 9.82E−08 |
| | | | Invasive ductal breast carcinoma epithelia (9) | −2.272 | −7.553 | 4.99E−07 |
| | | | Ductal breast carcinoma in situ stroma (9) | −2.008 | −4.236 | 1.60E−04 |
| | Zhao | Breast (3) | Invasive ductal breast carcinoma (37) | −4.418 | −8.645 | 1.20E−08 |
| | | | Lobular breast carcinoma (21) | −3.452 | −7.496 | 7.44E−07 |
| | Turashvili | Ductal breast cell (10) | Invasive lobular breast carcinoma (5) | −8.342 | −8.342 | 0.002 |
| | | Bular breast cell (10) | | | | |
| | Sorlie | Breast (4) | Ductal breast carcinoma (63) | −2.554 | −5.64 | 2.42E−04 |
| | Richardson | Breast (7) | Ductal breast carcinoma (40) | −4.237 | −7.552 | 9.00E−10 |
| | Sorlie 2 | Breast (4) | Ductal breast carcinoma (86) | −2.501 | −5.143 | 0.002 |
| | Perou | Breast (3) | Ductal breast carcinoma (35) | −2.535 | −4.175 | 0.005 |
| | Gluck | Breast (4) | Invasive breast carcinoma (154) | −2.041 | −7.529 | 2.46E−05 |
| | Finak | Breast (6) | Invasive breast carcinoma (53) | 8.451 | 13.331 | 1.59E−13 |
| NFIC | Richardson | Breast (7) | Ductal breast carcinoma (40) | −2.196 | −5.431 | 6.11E−06 |
| | Sorlie 2 | Breast (4) | Lobular breast carcinoma (6) | −2.029 | −3.852 | 0.003 |

| Gene | Dataset | Normal (cases) | Tssumor (cases) | Fold change | t-Test | p-Value |
|------|---------|----------------|-----------------|-------------|--------|---------|
| NFIX | Curtis | Breast (144) | Invasive ductal breast carcinoma (1,556) | −2.138 | −27.969 | 2.43E−104 |
| | | | Invasive ductal and invasive lobular breast carcinoma (90) | −2.121 | −12.042 | 5.06E−22 |
| | Sorlie | Breast (4) | Ductal breast carcinoma (63) | −2.417 | −6.071 | 5.29E−05 |
| | Perou | Breast (3) | Ductal breast carcinoma (36) | −2.292 | −5.401 | 6.13E−04 |
| | Sorlie 2 | Breast (4) | Ductal breast carcinoma (92) | −2.319 | −5.898 | 4.21E−04 |
| | Zhao | Breast (3) | Invasive ductal breast carcinoma (34) | −2.214 | −6.293 | 2.85E−06 |
| | Karnoub | Breast (15) | Invasive ductal breast carcinoma stroma (7) | −2.012 | −3.55 | 0.001 |
| | Gluck | Breast (4) | Invasive breast carcinoma (154) | −2.573 | −7 | 5.94E−04 |

**Note:**
NFI, Nuclear factor I; TCGA, The Cancer Genome Atlas.

conducted (Fig. S1). Breast invasive carcinoma patients with a NFIX gene alteration showed significantly poor overall survival (OS) and disease-free survival (DFS) compared with breast invasive carcinoma patients without NFIX gene alteration. In addition, the correlation between NFI gene expression and its DNA methylation was calculated using the cBioPortal online tool for breast invasive carcinoma (TCGA, Firehose Legacy), and Pearson's correction was included (Figs. 2D–2G). The results indicated significant and negative correlations between NFI gene expression and corresponding DNA methylation in breast invasive carcinoma. Regression analysis confirmed a strongly negative correlation in NFIA (Pearson's $r = -0.64$) and NFIX (Pearson's $r = -0.76$), a moderately negative correlation in NFIB (Pearson's $r = -0.41$), and a weakly negative correlation in NFIC (Pearson's $r = -0.30$).

Subsequently, the prognostic effects of the NFI family members were determined using the KM plotter database (www.kmplot.com) (*Gyorffy et al., 2010*). The breast oncology community currently describes breast cancer in terms of intrinsic biologic subtypes, and at least four subtypes are defined: basal-like (ER−/PR−/HER2−), luminal A (ER+/HER2−/grade 1 or 2), luminal B (ER+/HER2−/grade 3) and HER2 enriched (any HER2+ tumor). Therefore, prognosis analysis was investigated based on these four intrinsic subtypes. The results showed decreased NFIA, NFIC and NFIX expression predicted worse recurrence-free survival (RFS) in all patient subtypes. In addition, low NFIA expression was associated with poor RFS, OS and distant metastasis-free survival (DMFS) in luminal A subtype. Decreased NFIA expression showed better RFS, OS and DMFS in the HER2-enriched subtype. Similarly, reduced NFIX expression indicated worse RFS in patients classified as luminal A but not in patients classified as HER2-enriched. In addition, downregulated NFIB expression was associated with better DMFS and post-progression survival (PPS) in basal-like patients. All the results are summarized in Table 2 and Tables S1A–S1C File.

### Transcription levels and prognostic significance of the NFI family members in lung cancer

Similarly, the Oncomine database was utilized to compare the mRNA expression levels of the NFI family members in lung cancer and normal tissues. Using the same thresholds

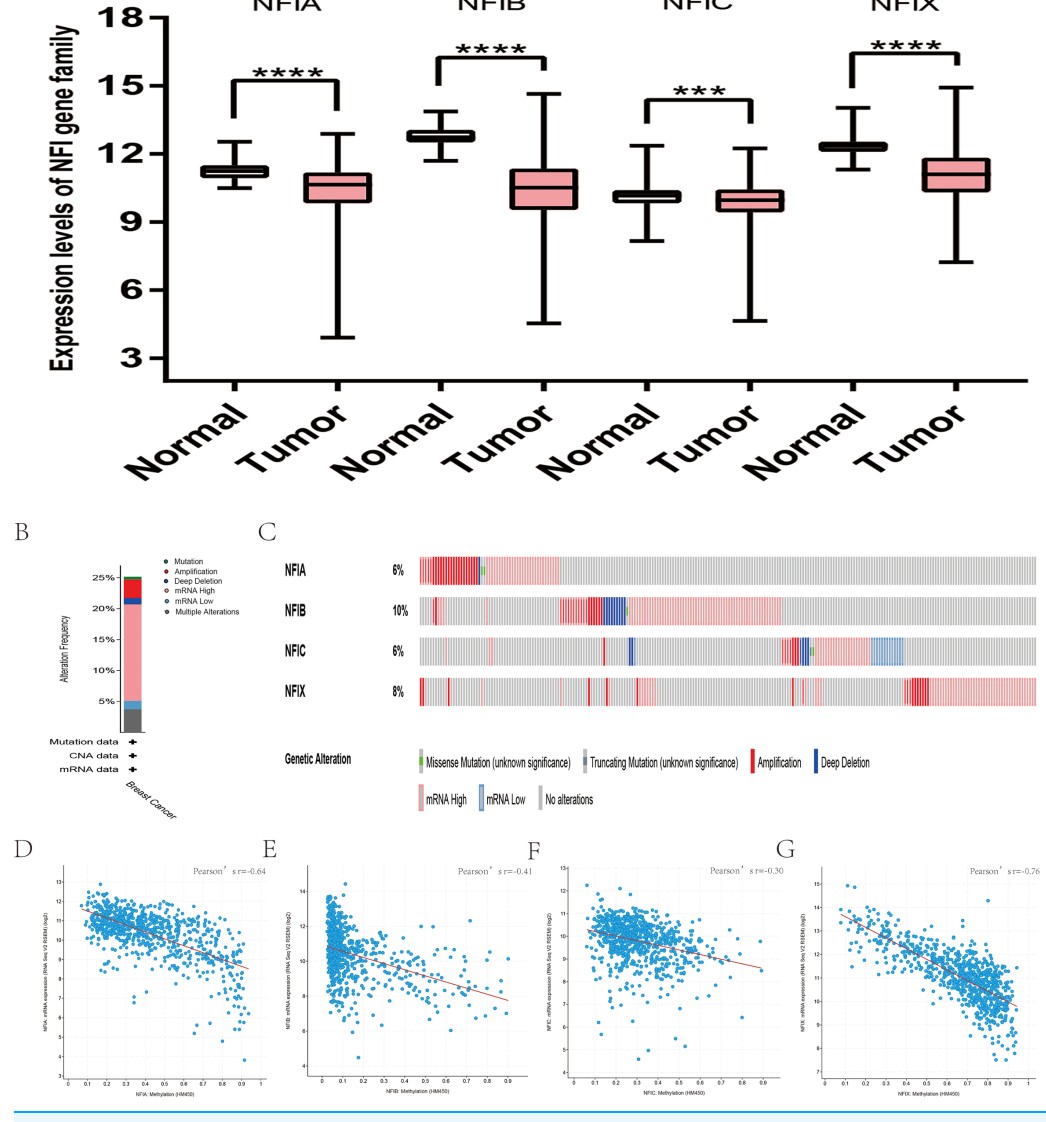

**Figure 2 Transcription levels, genomic analysis of the NFIs and regression analysis between the mRNA expression of the NFIs and its corresponding methylation in breast cancer (A–G).** Box-whisker plots show the differences in transcript levels of the NFI family members between normal and tumors samples. The median value is represented by the middle line in the boxes. Statistical differences were examined by two-tailed student's $t$-test. ***$p < 0.001$, ****$p < 0.0001$. NFI, Nuclear factor I; TCGA, The Cancer Genome Atlas.

($p$-value = 0.01; fold change = 2; gene rank: 10%, data type: mRNA), in Okayama and Hou's analysis, the NFIA mRNA expression level was significantly lower in lung adenocarcinoma (*Hou et al., 2010*; *Okayama et al., 2012*). According to Garber's database, NFIA expression was also downregulated in lung adenocarcinoma and large cell lung carcinoma (*Garber et al., 2001*). However, in SCLC and squamous cell lung carcinoma, opposite results were observed for NFIA in reporter IMAGE:364302 (high expression) and IMAGE:813154 (low expression). The NFIB mRNA expression level was significantly decreased in seven unique analyses, across different lung cancer subtypes, in five different

**Table 2 Correlation of NFIs with survival outcomes in breast cancer patients.**

| Gene | Affymetrix ID | Survival outcome | No. of cases | Cut-off value | HR | 95% CI | *p*-Value |
|------|---------------|------------------|--------------|---------------|-----|--------|-----------|
| NFIA | 226806_s_at | OS | 1,402 | 1,490 | 0.9 | [0.65–1.22] | 0.49 |
|      |             | RFS | 3,951 | 1,519 | 0.72 | [0.61–0.84] | <0.001 |
|      |             | DMFS | 1,746 | 1,484 | 0.94 | [0.68–1.3] | 0.72 |
|      |             | PPS | 414 | 1,488 | 1.03 | [0.72–1.47] | 0.88 |
| NFIB | 213029_at | OS | 1,402 | 1,292 | 0.94 | [0.76–1.16] | 0.56 |
|      |           | RFS | 3,951 | 1,216 | 1.1 | [0.99–1.23] | 0.086 |
|      |           | DMFS | 1,746 | 1,226 | 1.09 | [0.9–1.32] | 0.39 |
|      |           | PPS | 414 | 1,277 | 1.2 | [0.94–1.53] | 0.14 |
| NFIC | 226377_at | OS | 1,402 | 690 | 0.79 | [0.58–1.08] | 0.14 |
|      |           | RFS | 3,951 | 1,155 | 0.84 | [0.72–0.98] | 0.025 |
|      |           | DMFS | 1,746 | 1,005 | 0.85 | [0.61–1.17] | 0.31 |
|      |           | PPS | 414 | 657 | 1.36 | [0.96–1.94] | 0.086 |
| NFIX | 227400_at | OS | 1,402 | 585 | 0.71 | [0.52–0.97] | 0.033 |
|      |           | RFS | 3,951 | 723 | 0.82 | [0.7–0.95] | 0.0098 |
|      |           | DMFS | 1,746 | 684 | 0.96 | [0.7–1.33] | 0.81 |
|      |           | PPS | 414 | 539 | 0.92 | [0.65–1.32] | 0.67 |

**Note:**
HR, hazard ratio; CI, confidence interval; OS, overall survival; RFS, progression free survival; DMFS, distant metastasis free survival; PPS, post progression survival.

databases including Stearman (*Stearman et al., 2005*), Bhattacharjee (*Bhattacharjee et al., 2001*), Garber (*Garber et al., 2001*), Wachi and Hou (*Hou et al., 2010*; *Wachi, Yoneda & Wu, 2005*). Two comparisons with Bhattacharjee's database indicated the NFIC mRNA level was reduced in lung adenocarcinoma and SCLC (*Bhattacharjee et al., 2001*). NFIX expression was also lower in large cell lung carcinoma according to Garber's database (*Garber et al., 2001*). All statistically significant results are shown in Table 3. Next, the expression differences between lung cancer and normal tissues were evaluated using the mRNA HiSeq expression data from the TCGA database. There were 110 normal samples and 1,019 lung cancer samples, including 513 lung adenocarcinomas and 506 lung squamous cell carcinomas. As shown in Fig. 3A, the expression of the NFI family members in lung cancer tissues was significantly lower than in normal tissues. Next, the underlying mechanism of dysregulated expression of the NFI family was analyzed using the cBioPortal online tool for lung adenocarcinoma (TCGA, Firehose Legacy) and lung squamous cell carcinoma (TCGA, Firehose Legacy). NFI genes were altered in 58 samples of 230 patients (25%) with lung adenocarcinoma and 41 samples of 178 patients (23%) with lung squamous cell carcinoma. The NFI genes in lung cancer were analyzed and depicted as oncoprints representing mutation, amplification, deep deletion, mRNA high, mRNA low, and multiple alterations (Figs. 3B, 3C, 3H and 3I). Survival analysis of the NFI genes with and without each gene alteration was conducted. Lung adenocarcinoma patients with NFIB gene alteration showed better DFS compared with lung adenocarcinoma patients without NFIB gene alteration (Fig. S2). Lung squamous cell carcinoma patients with NFIA gene alteration showed worse OS compared with lung

**Table 3 Datasets of the NFI family in lung cancer (ONCOMINE database).**

| Gene | Dataset | Normal (cases) | Tumor (cases) | Fold change | *t*-Test | *p*-Value |
|------|---------|----------------|---------------|-------------|----------|-----------|
| NFIA | Okayama | Lung (20) | Lung adenocarcinoma (226) | −2.25 | −15.234 | 1.13E−25 |
| | Garber | Lung (5) | Lung adenocarcinoma (40) | −2.404 | −7.113 | 3.60E−06 |
| | | Lung (5) | Large cell lung carcinoma (4) | −2.498 | −5.019 | 0.002 |
| | | Lung (5) | Small cell lung carcinoma (4) | −2.657 | −5.127 | 0.002 |
| | | Lung (5) | Squamous cell lung carcinoma (13) | −2.325 | −3.934 | 5.77E−04 |
| | | Lung (5) | Small cell lung carcinoma (4) | 3.708 | 4.848 | 7.78E−04 |
| | | Lung (5) | Squamous cell lung carcinoma (13) | 4.124 | 4.747 | 2.93E−04 |
| | Hou | Lung (65) | Lung adenocarcinoma (45) | −2.564 | −7.923 | 4.56E−11 |
| NFIB | Stearman | Lung (19) | Lung adenocarcinoma (20) | −2.163 | −8.091 | 2.63E−09 |
| | Bhattacharjee | Lung (17) | Lung carcinoid tumor (20) | −20.662 | −8.304 | 6.88E−10 |
| | | | Lung adenocarcinoma (132) | −4.608 | −5.259 | 2.21E−05 |
| | | | Squamous cell lung carcinoma (21) | −3.994 | −3.58 | 5.13E−04 |
| | Garber | Lung (5) | Large cell lung carcinoma (4) | −2.177 | −4.062 | 0.008 |
| | Wachi | Lung (5) | Squamous cell lung carcinoma (5) | −2.432 | −3.578 | 0.004 |
| | Hou | Lung (65) | Squamous cell lung carcinoma (27) | −2.422 | −9.841 | 3.81E−12 |
| NFIC | Bhattacharjee | Lung (17) | Lung adenocarcinoma (132) | −4.007 | −4.325 | 2.01E−04 |
| | | | Small cell lung carcinoma (6) | −4.793 | −4.575 | 8.63E−05 |
| NFIX | Garber | Lung (5) | Large cell lung carcinoma (4) | −2.032 | −4.228 | 0.003 |

squamous cell carcinoma patients without NFIA gene alterations (Fig. S3). In addition, the correlation between NFI gene expression and its DNA methylation was calculated using the cBioPortal online tool for lung adenocarcinoma (TCGA, Firehose Legacy) and lung squamous cell carcinoma (TCGA, Firehose Legacy), and Pearson's correction was included. The results indicated significant and negative correlation between NFI gene expression and corresponding DNA methylation in lung adenocarcinoma (Figs. 3D–3G) and lung squamous cell carcinoma (Figs. 3J–3M). Regarding lung adenocarcinoma, regression analysis confirmed a strongly negative correlation in NFIX (Pearson's $r = -0.63$), a moderately negative correlation in NFIA (Pearson's $r = -0.47$) and NFIB (Pearson's $r = -0.45$), and a weakly negative correlation in NFIC (Pearson's $r = -0.21$). Regression analysis indicated a moderately negative correlation in NFIB (Pearson's $r = -0.53$) and NFIX (Pearson's $r = -0.42$) and a weakly negative correlation in NFIA (Pearson's $r = -0.38$). However, correlation between NFIC expression and methylation was not observed.

Next, the prognostic value of the NFI family members was assessed for lung cancer using the KM plotter database (*Gyorffy et al., 2013*). OS, first progression (FP), and PPS were analyzed for each gene. NFIC was uncorrelated with OS, FP and PPS in patients with lung adenocarcinoma and squamous cell lung carcinoma. Decreased NFIA and NFIB expression predicted worse OS and FP in lung adenocarcinoma patients. Reduced NFIB expression was also associated with poor PPS. No gene showed statistical significance for squamous cell lung carcinoma patients except NFIX, which was associated with OS. All the detailed prognostic analyses are shown in Table 4 and in Tables S2A and S2B File.

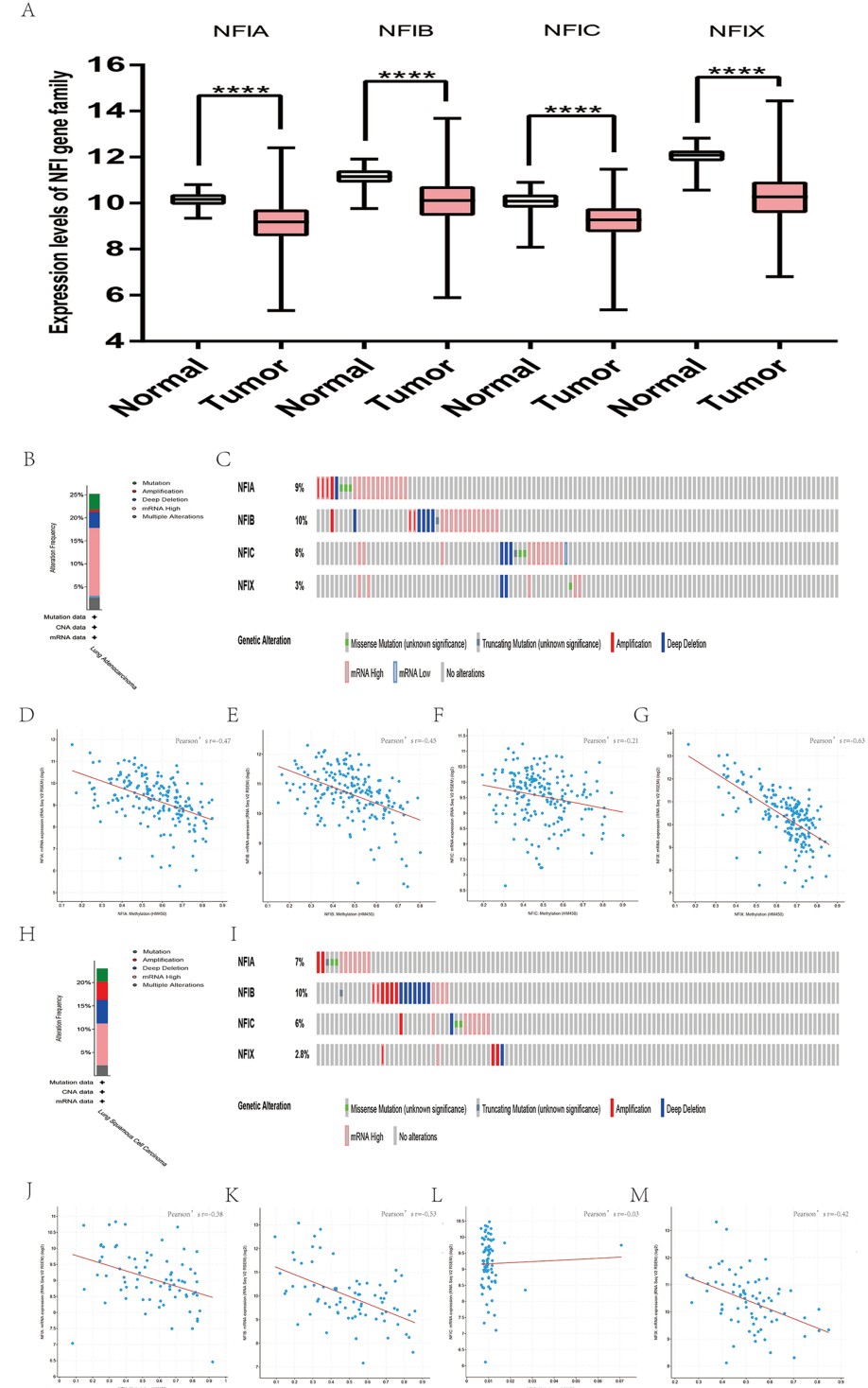

**Figure 3 Transcription levels, genomic analysis of the NFIs and regression analysis between the mRNA expression of the NFIs and its corresponding methylation in lung cancer (A–M).** Box-whisker plots show the differences in transcript levels of the NFI family members between normal and tumors samples. The median value is represented by the middle line in the boxes. Statistical differences were examined by two-tailed student's *t*-test. ****$p < 0.0001$. NFI, Nuclear factor I; TCGA, The Cancer Genome Atlas.

**Table 4 Correlation of NFIs with survival outcomes in lung cancer patients.**

| Gene | Affymetrix ID | Survival outcome | No. of cases | Cut-off value | HR | 95% CI | *p*-Value |
|------|---------------|------------------|--------------|---------------|----|--------|-----------|
| NFIA | 226806_s_at | OS | 1,926 | 697 | 0.58 | [0.49–0.69] | <0.001 |
|      |             | FP | 982 | 981 | 0.69 | [0.53–0.9] | 0.0069 |
|      |             | PPS | 344 | 1,015 | 0.59 | [0.38–0.91] | 0.016 |
| NFIB | 213029_at | OS | 1,926 | 1,014 | 0.69 | [0.61–0.78] | <0.001 |
|      |           | FP | 982 | 1,179 | 0.9 | [0.74–1.09] | 0.28 |
|      |           | PPS | 344 | 1,237 | 0.66 | [0.52–0.86] | 0.0015 |
| NFIC | 226377_at | OS | 1,926 | 439 | 0.91 | [0.77–1.08] | 0.27 |
|      |           | FP | 982 | 464 | 1.16 | [0.88–1.51] | 0.29 |
|      |           | PPS | 344 | 426 | 1.7 | [1.1–2.62] | 0.016 |
| NFIX | 227400_at | OS | 1,926 | 339 | 0.97 | [0.82–1.14] | 0.72 |
|      |           | FP | 982 | 373 | 1.2 | [0.92–1.57] | 0.18 |
|      |           | PPS | 344 | 353 | 1.18 | [0.77–1.82] | 0.44 |

**Note:**
HR, hazard ratio; CI, confidence interval; OS, overall survival; FP, first progression; PPS, post progression survival.

**Table 5 Datasets of the NFI family in bladder cancer (ONCOMINE database).**

| Gene | Dataset | Normal (cases) | Tumor (cases) | Fold change | *t*-Test | *p*-Value |
|------|---------|----------------|---------------|-------------|----------|-----------|
| NFIA | Lee | Bladder mucosa (68) | Superficial bladder cancer (126) | −2.034 | −7.839 | 8.56E−13 |
|      |     |                     | Infiltrating bladder urothelial carcinoma (62) | −2.02 | −5.127 | 5.30E−07 |
| NFIB | Sanchez-Carbayo 2 | Bladder (48) | Infiltrating bladder urothelial carcinoma (81) | −3.224 | −11.267 | 8.55E−21 |
|      |                   |              | Superficial bladder cancer (28) | −5.44 | −10.279 | 3.85E−13 |
|      | Lee | Bladder mucosa (68) | Superficial bladder cancer (126) | −2.817 | −8.17 | 9.13E−14 |
| NFIC | Blaveri 2 | Bladder (3) | Superficial bladder cancer (26) | −3.478 | −16.225 | 2.22E−15 |
|      |           |             | Infiltrating bladder urothelial carcinoma (53) | −2.182 | −13.966 | 7.22E−20 |
|      | Sanchez-Carbayo 2 | Bladder (48) | Superficial bladder cancer (28) | −3.362 | −5.985 | 8.05E−08 |
| NFIX | Lee | Bladder mucosa (68) | Superficial bladder cancer (126) | −2.417 | −8.103 | 9.27E−14 |

## Transcription levels and prognostic significance of the NFI family members in bladder cancer

For bladder cancer, all statistically significant datasets from the Oncomine database were extracted in Table 5. NFIA, NFIB and NFIC expressions were decreased in both superficial bladder cancer and infiltrating bladder urothelial carcinoma compared with normal tissues in Lee (*Lee et al., 2010*), Sanchez-Carbayo2 and Blaveri2's studies (*Blaveri et al., 2005*; *Sanchez-Carbayo et al., 2006*). According to Lee's study, NFIX expression was reduced in superficial bladder cancer (*Lee et al., 2010*). Based on mRNA HiSeq expression data from the TCGA database, the expression of all NFI family members was significantly downregulated in 407 bladder cancer samples compared with 19 normal samples (Fig. 4A). Next, the cBioPortal online tool was used to investigate the underlying mechanism of dysregulated expression of the NFI family for bladder urothelial carcinoma (TCGA, Firehose Legacy). NFI genes were altered in 38 samples of 127 patients (30%)

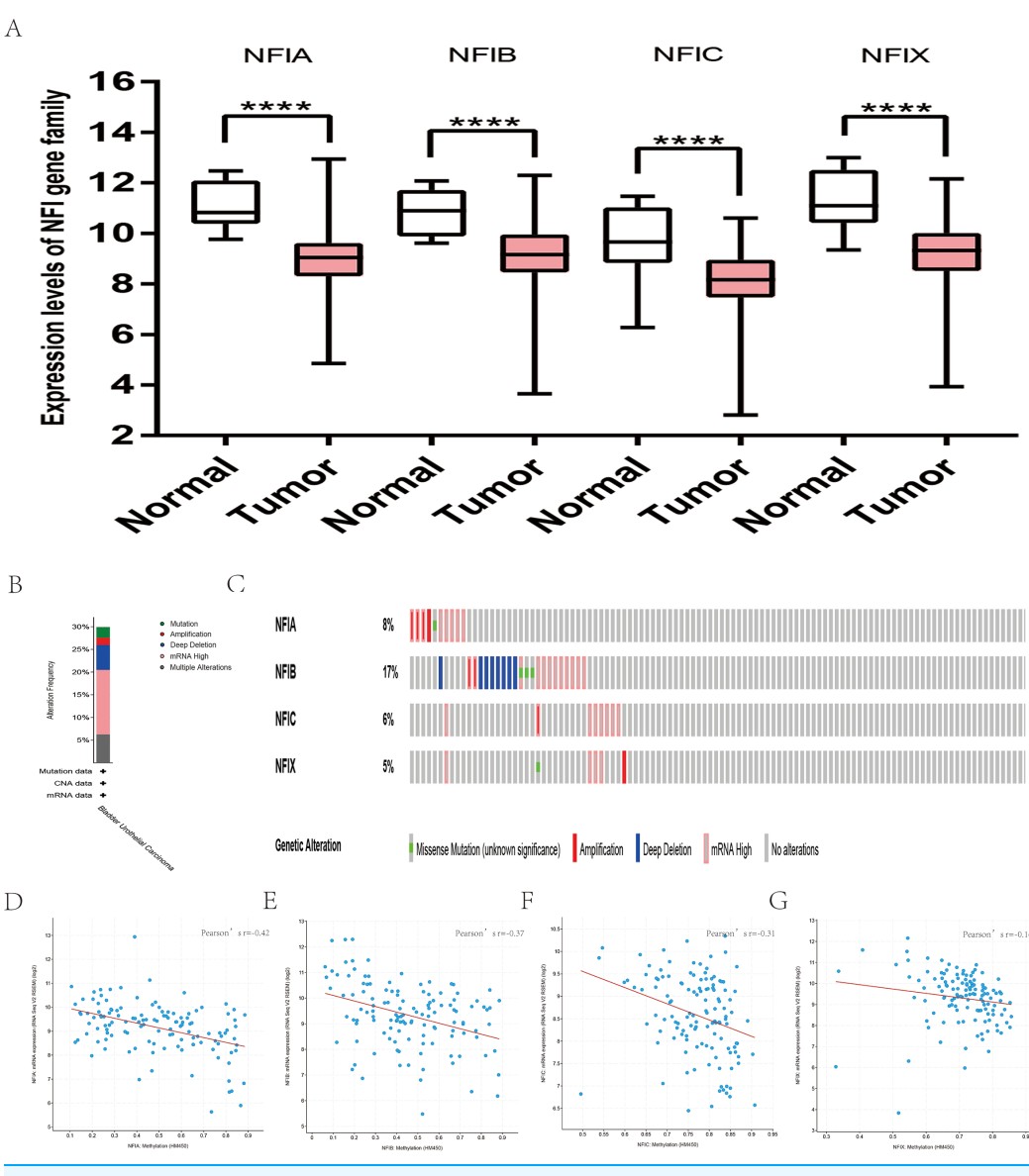

**Figure 4 Transcription levels, genomic analysis of the NFIs and regression analysis between the mRNA expression of the NFIs and its corresponding methylation in bladder cancer (A–G).** Box-whisker plots show the differences in transcript levels of the NFI family members between normal and tumors samples. The median value is represented by the middle line in the boxes. Statistical differences were examined by two-tailed student's *t*-test. ****$p < 0.0001$. NFI, Nuclear factor I; TCGA, The Cancer Genome Atlas.

with bladder urothelial carcinoma. Genetic alteration of the NFI genes was analyzed and depicted as oncoprints representing mutation, amplification, deep deletion, mRNA high, and multiple alterations (Figs. 4B and 4C). Survival analysis of the NFI genes with and without each gene alteration was conducted (Fig. S4). Bladder urothelial carcinoma patients with NFIB gene alteration showed significantly better OS compared with bladder urothelial patients without NFIB gene alteration. In addition, the correlation between NFI gene expression and its DNA methylation was calculated using the cBioPortal online tool for bladder urothelial carcinoma (TCGA, Firehose Legacy), and Pearson's correction

**Table 6 Correlation of NFIs with survival outcomes in bladder cancer patients.**

| Gene | RNAseq ID | Survival outcome | No. of cases | Cut-off value | HR | 95% CI | *p*-Value |
|------|-----------|------------------|--------------|---------------|------|------------|-----------|
| NFIA | 4774 | OS | 404 | 662 | 1.72 | [1.28–2.31] | <0.001 |
|      |      | RFS | 187 | 804 | 0.54 | [0.22–1.31] | 0.16 |
| NFIB | 4781 | OS | 404 | 767 | 0.87 | [0.65–1.16] | 0.34 |
|      |      | RFS | 187 | 591 | 2.19 | [0.98–4.9] | 0.051 |
| NFIC | 4782 | OS | 404 | 2,028 | 1.59 | [1.18–2.13] | 0.0018 |
|      |      | RFS | 187 | 1,858 | 1.73 | [0.84–4.9] | 0.13 |
| NFIX | 4784 | OS | 404 | 1,176 | 1.36 | [1.01–1.83] | 0.045 |
|      |      | RFS | 187 | 419 | 1.68 | [0.64–4.37] | 0.29 |

Note:
HR, hazard ratio; CI, confidence interval; OS, overall survival; RFS, relapse free survival.

was included (Figs. 4D–4G). The results indicated significantly negative correlation between NFI gene expression and corresponding DNA methylation in bladder urothelial carcinoma. Regression analysis confirmed a moderately negative correlation in NFIA (Pearson's $r = -0.42$) and a weakly negative correlation in NFIB (Pearson's $r = -0.37$) and NFIC (Pearson's $r = -0.31$). However, correlation between NFIX expression and its methylation was not observed (Pearson's $r = -0.16$).

Subsequently, the association between the NFI family members and the survival outcomes of bladder cancer patients using the KM plotter database was explored (*Nagy et al., 2018*). High expression of NFIA, NFIC and NFIX predicted worse survival outcome in patients with bladder cancer. All the data are shown in Table 6.

## Transcription levels and prognostic significance of the NFI family members in head and neck cancer

For head and neck cancer, a total of six datasets from the Oncomine database were used to investigate the mRNA expression of the NFI family members in tumors and normal tissues (Table 7). Ye's dataset showed significantly decreased NFIA and NFIB mRNA expression level in tongue squamous cell carcinoma (*Ye et al., 2008*). In addition, expression of NFIB and NFIX was downregulated in tongue squamous cell carcinoma according to Estilo's study (*Estilo et al., 2009*). NFIA and NFIX were reduced in tonsillar carcinoma, nasopharyngeal carcinoma, and oral cavity squamous cell carcinoma, respectively. In Cromer's dataset, NFIB mRNA was significantly decreased in head and neck squamous cell carcinoma (*Cromer et al., 2004*). Nevertheless, the NFIB mRNA expression level was significantly elevated in salivary gland adenoid cystic carcinoma in FriersonHF's dataset (*Frierson et al., 2002*). Due to the limited number of cases in the Oncomine database, 522 head and neck squamous cell carcinomas and 44 normal samples from the TCGA database were further used to validate the potential expression differences of the NFI family members in tumors and normal tissues. Expression of all the NFI family members was significantly downregulated in head and neck squamous cell carcinoma compared with normal tissues (Fig. 5A). Next, the underlying mechanism of dysregulated expression of the NFI family was investigated using the cBioPortal online tool

**Table 7 Datasets of the NFI family in head-neck cancer (ONCOMINE database).**

| Gene | Dataset | Normal (cases) | Tumor (cases) | Fold change | t-Test | p-Value |
|------|---------|----------------|---------------|-------------|--------|---------|
| NFIA | Ye head–neck | Normal (Tongue (12)) | Tongue squamous cell carcinoma (26) | −3.296 | −7.117 | 2.53E−08 |
| | Pyeon multi-cancer | Normal (Cervix Uteri (8) Oral Cavity (9) Palate (1) Tonsil (4)) | Tonsillar carcinoma (6) | −2.285 | −4.799 | 3.09E−05 |
| | Sengupta head–neck | Normal (Nasopharynx (10)) | Nasopharyngeal carcinoma (31) | −2.036 | −4.997 | 5.60E−05 |
| NFIB | Estilo head–neck | Normal (Tongue (26)) | Tongue squamous cell carcinoma (31) | −3.388 | −8.191 | 3.28E−11 |
| | Ye head–neck | Normal (Tongue (12)) | Tongue squamous cell carcinoma (26) | −2.984 | −3.941 | 1.91E−04 |
| | Cromer head–neck | Normal (Uvula (4)) | Head and neck squamous cell carcinoma (34) | −2.63 | −4.165 | 5.86E−04 |
| | FriersonHF salivary-gland | Normal (Salivary gland (6)) | Salivary gland adenoid cystic carcinoma (16) | 2.485 | 6.724 | 1.21E−06 |
| NFIX | Peng head–neck | Normal (Oral cavity (22)) | Oral cavity squamous cell carcinoma (57) | −2.754 | −19.426 | 1.17E−31 |
| | Estilo head–neck | Normal (Tongue (26)) | Tongue squamous cell carcinoma (31) | −2.343 | −3.731 | 2.90E−04 |

for head and neck squamous cell carcinoma (TCGA, Firehose Legacy). NFIs were altered in 106 samples of 504 patients (21%) with head and neck squamous cell carcinoma. Specifically, genetic alteration of the NFI genes was analyzed and depicted as oncoprints representing mutation, amplification, deep deletion, mRNA high and multiple alterations (Figs. 5B and 5C). Survival analysis of the NFI genes with and without each gene alteration was conducted (Fig. S5). Head and neck cancer patients with NFIA gene alteration showed better OS compared with head and neck cancer patients without NFIA gene alteration. In addition, the correlation between NFI gene expression and its DNA methylation was calculated using the cBioPortal online tool for head and neck squamous cell carcinoma (TCGA, Firehose Legacy), and Pearson's correction was included (Figs. 5D–5G). The results indicated significantly negative correlation between NFI gene expression and corresponding DNA methylation in head and neck squamous cell carcinoma. Regression analysis confirmed a moderately negative correlation in NFIB (Pearson's $r = −0.41$) and a weakly negative correlation in NFIA (Pearson's $r = −0.33$) and NFIX (Pearson's $r = −0.36$). Correlation between NFIC expression and its methylation was not observed (Pearson's $r = −0.19$). However, only survival data of head and neck squamous cell carcinoma from the KM plotter database were analyzed (*Nagy et al., 2018*). The results showed all the NFI family members were associated with OS in head and neck squamous cell carcinoma except NFIX. Higher expression levels of NFIA, NFIB and NFIC implied better OS (Table 8).

## Transcription levels and prognostic significance of the NFI family members in esophageal cancer

All statistically significant datasets for esophageal cancer were extracted in Table 9. NFIA was reduced in esophageal squamous cell carcinoma (ESCC), esophageal adenocarcinoma, and Barrett's esophagus in Su2 and Kim's datasets (*Kim et al., 2010*; *Su et al., 2011*). In Kim's study, the NFIB mRNA expression level was also downregulated in esophageal
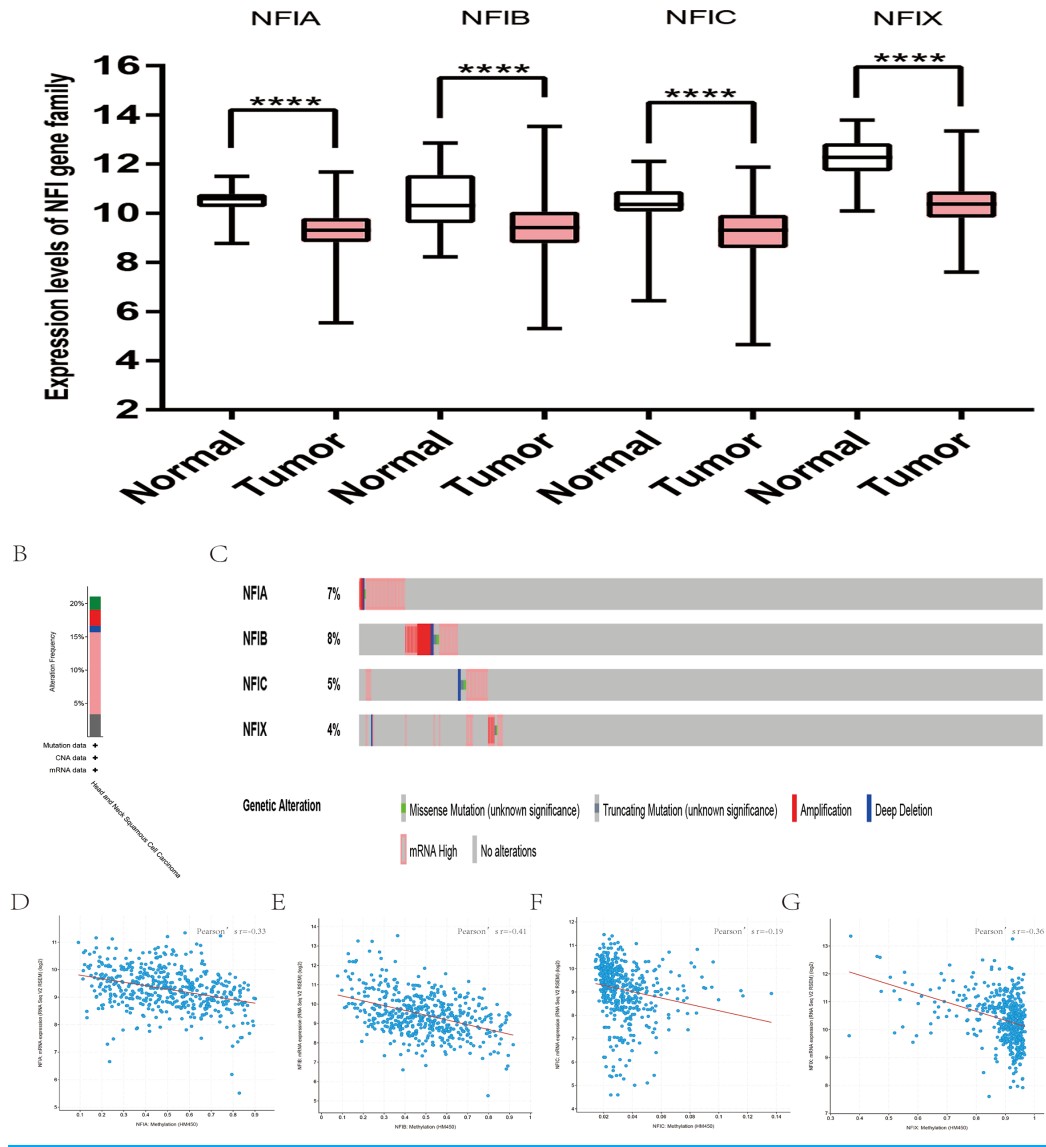

**Figure 5 Transcription levels, genomic analysis of the NFIs and regression analysis between the mRNA expression of the NFIs and its corresponding methylation in head-neck cancer (A–G).** Box-whisker plots show the differences in transcript levels of the NFI family members between normal and tumors samples. The median value is represented by the middle line in the boxes. Statistical differences were examined by two-tailed student's *t*-test. ****$p < 0.0001$. NFI, Nuclear factor I; TCGA, The Cancer Genome Atlas.

adenocarcinoma and Barrett's esophagus. However, Hao's dataset showed the opposite results for esophageal adenocarcinoma and Barrett's esophagus (*Hao et al., 2006*).

In the TCGA database, the expression of the NFI family members was decreased in esophageal cancer evaluated using the mRNA HiSeq expression data (Fig. 6A). Next, the underlying mechanism of dysregulated expression of the NFI family was investigated using the cBioPortal online tool for esophageal carcinoma (TCGA, Firehose Legacy). NFI genes were altered in 38 samples of 184 patients (21%) with esophageal carcinoma.

**Table 8 Correlation of NFIs with survival outcomes in head-neck cancer patients.**

| Gene | RNAseq ID | Survival outcome | No. of cases | Cut-off value | HR | 95% CI | *p*-Value |
|------|-----------|------------------|--------------|---------------|-----|--------|-----------|
| NFIA | 4774 | OS | 499 | 838 | 0.6 | [0.45–0.81] | <0.001 |
|      |      | RFS | 124 | 827 | 0.53 | [0.24–1.17] | 0.11 |
| NFIB | 4781 | OS | 499 | 487 | 0.74 | [0.55–0.99] | 0.043 |
|      |      | RFS | 124 | 1,288 | 0.56 | [0.21–1.47] | 0.23 |
| NFIC | 4782 | OS | 499 | 1,705 | 0.64 | [0.48–0.85] | 0.0019 |
|      |      | RFS | 124 | 3,021 | 0.64 | [0.24–1.7] | 0.37 |
| NFIX | 4784 | OS | 499 | 1,959 | 0.76 | [0.57–1.02] | 0.069 |
|      |      | RFS | 124 | 1,819 | 0.47 | [0.19–1.15] | 0.091 |

Note:
HR, hazard ratio; CI, confidence interval; OS, overall survival; RFS, relapse free survival.

**Table 9 Datasets of the NFI family in esophageal carcinoma (ONCOMINE database).**

| Gene | Dataset | Normal (cases) | Tumor (cases) | Fold change | *t*-Test | *p*-Value |
|------|---------|----------------|---------------|-------------|----------|-----------|
| NFIA | Su Esophagus 2 | Normal (Esophagus (51)) | Esophageal squamous cell carcinoma (51) | −2.142 | −9.685 | 3.66E−16 |
|      | Kim Esophagus | Normal (Esophagus (28)) | Esophageal adenocarcinoma (75) | −2.465 | −9.118 | 4.98E−15 |
|      |      |      | Barrett's esophagus (15) | −2.171 | −7.186 | 6.82E−08 |
| NFIX | Kim Esophagus | Normal (Esophagus (28)) | Esophageal adenocarcinoma (75) | −4.387 | −10.777 | 9.43E−19 |
|      |      |      | Barrett's esophagus (15) | −3.31 | −9.268 | 1.01E−09 |
|      | Hao Esophagus | Normal (Duodenum (13) Esophagus (15)) | Esophageal adenocarcinoma (5) | 2.447 | 4.617 | 4.96E−05 |
|      |      | Normal (Duodenum (11) Esophagus (13)) | Barrett's esophagus (12) | 2.364 | 3.701 | 5.09E−04 |

Genetic alteration of the NFI genes was analyzed and depicted as oncoprints representing mutation, amplification, deep deletion, mRNA high and multiple alterations (Figs. 6B and 6C). Survival analysis of the NFI genes with and without each gene alteration was conducted (Fig. S6). However, significant difference in survival was not observed. In addition, the correlation between NFI gene expression and its DNA methylation was calculated using the cBioPortal online tool for esophageal carcinoma (TCGA, Firehose Legacy), and Pearson's correction was included (Figs. 6D–6G). The results indicated significantly negative correlation between NFI gene expression and corresponding DNA methylation in esophageal carcinoma. Regression analysis confirmed a moderately negative correlation in NFIB (Pearson's $r = -0.4$) and NFIX (Pearson's $r = -0.51$), a weakly negative correlation in NFIA (Pearson's $r = -0.32$) and NFIC (Pearson's $r = -0.33$). The correlation between expression of the NFI family members and survival outcome involving OS in esophageal cancer patients was then determined using the KM plotter database. Low expression of NFIC and NFIX revealed poor prognosis in esophageal adenocarcinoma patients. The details are shown in Tables 10A and 10B.

## Transcription levels and prognostic significance of the NFI family members in kidney cancer

For kidney cancer, Higgins's dataset showed the NFIA mRNA expression level was upregulated in clear cell sarcoma of the kidney compared with normal kidney tissues

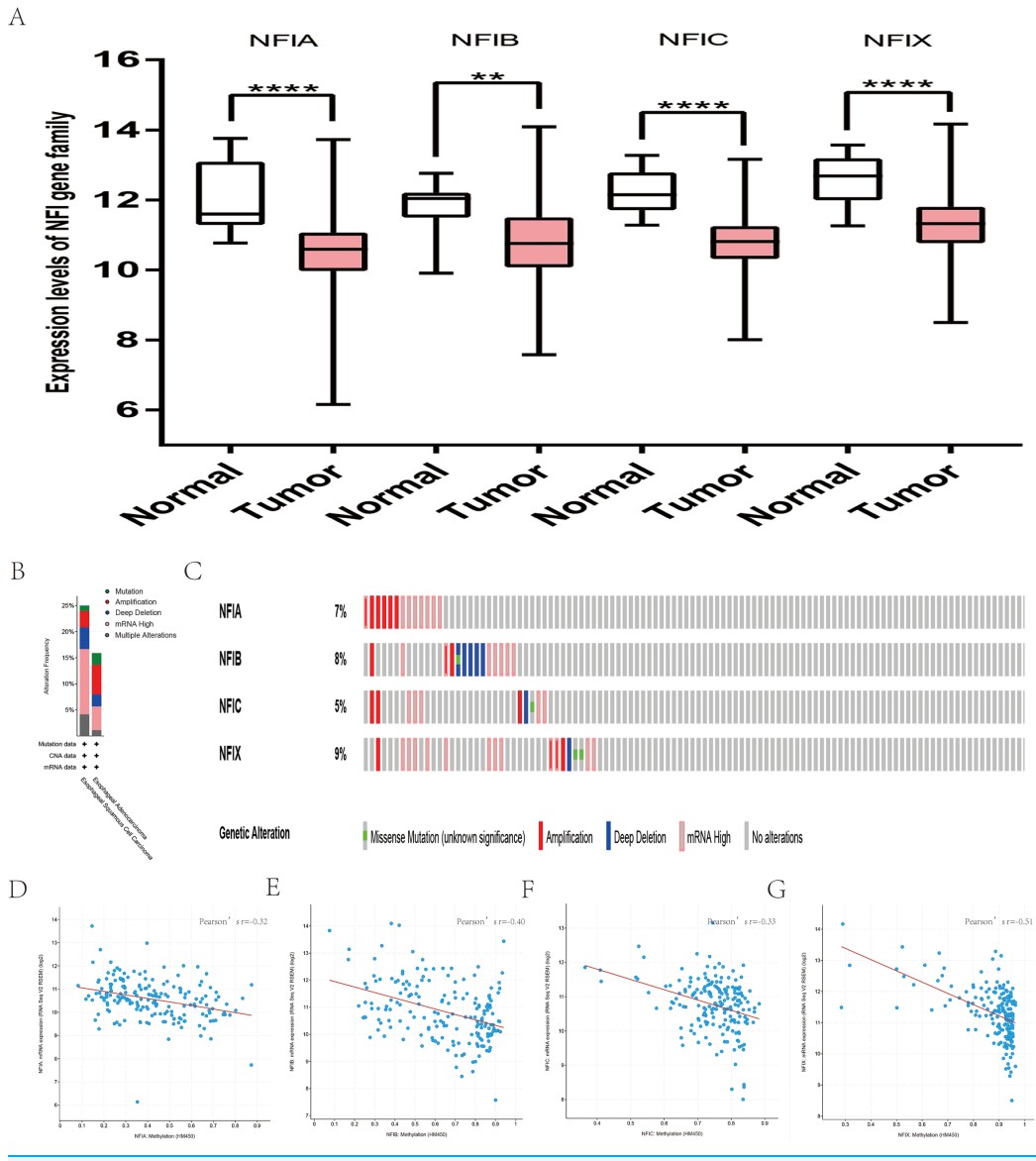

**Figure 6 Transcription levels, genomic analysis of the NFIs and regression analysis between the mRNA expression of the NFIs and its corresponding methylation in esophageal cancer (A–G).** Box-whisker plots show the differences in transcript levels of the NFI family members between normal and tumors samples. The median value is represented by the middle line in the boxes. Statistical differences were examined by two-tailed student's $t$-test. $**p < 0.01$, $****p < 0.0001$. NFI, Nuclear factor I; TCGA, The Cancer Genome Atlas.

(*Higgins et al., 2003*). NFIB was overexpressed in clear cell sarcoma of the kidney and papillary renal cell carcinoma according to Cutcliffe and Jones's datasets (*Cutcliffe et al., 2005*; *Jones et al., 2005*). However, in Cutcliffe's dataset, NFIB expression was reduced in Wilms tumor. In addition, two analyses with Yusenko's dataset showed lower NFIB mRNA levels in chromophobe renal cell carcinoma and renal oncocytoma (*Yusenko et al., 2009*). All statistically significant results are summarized in Table 11. Furthermore, analyses were performed for kidney chromophobe cell carcinoma (Fig. 7A), kidney clear

**Table 10 Correlation of NFIs with survival outcomes in esophageal cancer patients.**

| Gene | RNAseq ID | Survival outcome | No. of cases | Cut-off value | HR | 95% CI | p-Value |
|---|---|---|---|---|---|---|---|
| (A) Correlation of NFIs with survival outcomes in esophageal squamous cell carcinoma patients | | | | | | | |
| NFIA | 4774 | OS | 81 | 2,029 | 0.64 | [0.27–1.55] | 0.32 |
| | | RFS | 54 | 1,211 | 0.57 | [0.22–1.49] | 0.25 |
| NFIB | 4781 | OS | 81 | 2,102 | 0.51 | [0.21–1.24] | 0.13 |
| | | RFS | 54 | 2,422 | 0.31 | [0.09–1.1] | 0.056 |
| NFIC | 4782 | OS | 81 | 7,712 | 0.45 | [0.18–1.13] | 0.08 |
| | | RFS | 54 | 9,377 | 1.45 | [0.53–3.91] | 0.47 |
| NFIX | 4784 | OS | 81 | 4,135 | 0.72 | [0.3–1.73] | 0.46 |
| | | RFS | 54 | 4,135 | 0.35 | [0.1–1.24] | 0.09 |
| (B) Correlation of NFIs with survival outcomes in esophageal adenocarcinoma patients | | | | | | | |
| NFIA | 4774 | OS | 80 | 2,000 | 0.6 | [0.31–1.15] | 0.12 |
| | | RFS | 19 | 2,161 | 6.57 | [0.66–65.46] | 0.068 |
| NFIB | 4781 | OS | 80 | 2,571 | 1.46 | [0.76–2.79] | 0.25 |
| | | RFS | 19 | 4,327 | 3.03 | [0.42–21.68] | 0.25 |
| NFIC | 4782 | OS | 80 | 3,741 | 0.44 | [0.22–0.9] | 0.02 |
| | | RFS | 19 | 3,579 | 271493742 | [0–lnf.] | 0.28 |
| NFIX | 4784 | OS | 80 | 3,353 | 0.33 | [0.16–0.68] | 0.0017 |
| | | RFS | 19 | 3373 | 0 | [0–lnf.] | 0.018 |

**Note:**
HR, hazard ratio; CI, confidence interval; OS, overall survival; RFS, relapse free survival.

**Table 11 Datasets of the NFI family in kidney cancer (ONCOMINE database).**

| Gene | Dataset | Normal (cases) | Tumor (cases) | Fold change | t-Test | p-Value |
|---|---|---|---|---|---|---|
| NFIA | Higgins Renal | Normal (Kidney (3)) | Clear cell sarcoma of the kidney (25) | 2.061 | 4.058 | 0.0005 |
| NFIB | Cutcliffe Renal | Normal (Fetal kidney (3)) | Renal Wilms tumor (18) | −5.72 | −7.098 | 5.18E−07 |
| | Yusenko Renal | Normal (Fetal kidney (2) Kidney (3)) | Chromophobe renal cell carcinoma (4) | −2.281 | −4.333 | 0.002 |
| | | | Renal oncocytoma (4) | −3.229 | −4.26 | 0.003 |
| | Cutcliffe Renal | Normal (Fetal kidney (3)) | Clear cell sarcoma of the kidney (14) | 3.771 | 15.494 | 1.27E−09 |
| | Jones Renal | Normal (Kidney (23)) | Papillary renal cell carcinoma (11) | 2.661 | 6.739 | 5.14E−07 |

cell carcinoma (Fig. 8A), and kidney papillary cell carcinoma (Fig. 9A) using the mRNA HiSeq expression data from the TCGA database. Consistent with the trend observed in the Oncomine database, NFIA, NFIB and NFIX were significantly overexpressed in kidney clear cell carcinoma compared with normal kidney tissue. In addition, expression of NFIA and NFIB was downregulated in kidney chromophobe cell carcinoma, whereas expression of NFIC and NFIX was upregulated. Contrary to kidney chromophobe cell carcinoma, expression of NFIA and NFIB was upregulated in kidney papillary cell carcinoma and expression of NFIC and NFIX was reduced. Subsequently, the underlying mechanism of dysregulated expression of the NFI family was investigated using the cBioPortal online tool for kidney chromophobe cell carcinoma (TCGA, Firehose Legacy),

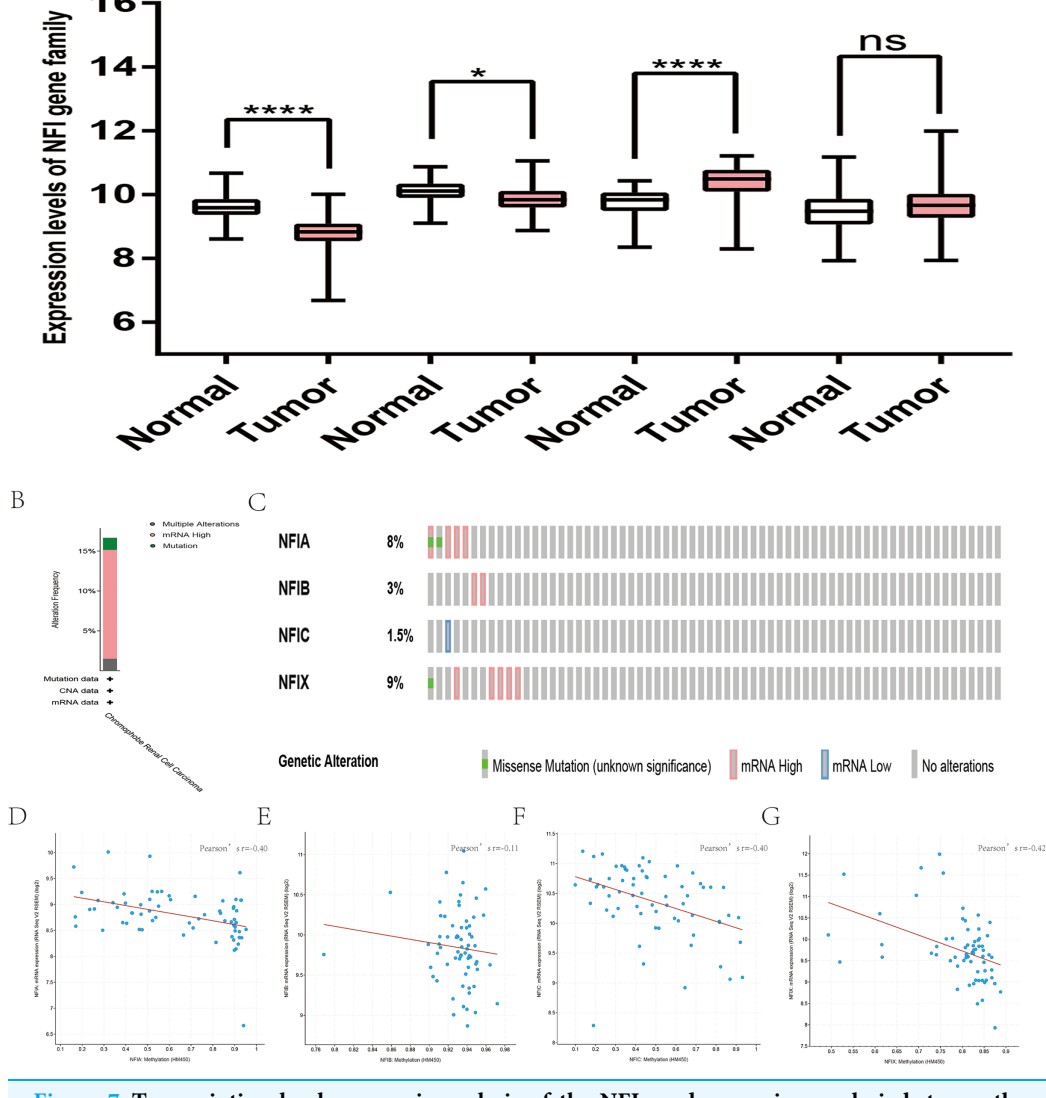

**Figure 7 Transcription levels, genomic analysis of the NFIs and regression analysis between the mRNA expression of the NFIs and its corresponding methylation in kidney chromophobe cell carcinoma (A–G).** Box-whisker plots show the differences in transcript levels of the NFI family members between normal and tumors samples. The median value is represented by the middle line in the boxes. Statistical differences were examined by two-tailed student's *t*-test. *$p < 0.05$, ****$p < 0.0001$, ns-non significant. NFI, Nuclear factor I; TCGA, The Cancer Genome Atlas.

kidney renal clear cell carcinoma (TCGA, Firehose Legacy) and kidney renal papillary cell carcinoma (TCGA, Firehose Legacy). NFI genes were altered in 11 samples of 66 patients (11%) with kidney chromophobe cell carcinoma, 72 samples of 448 patients (16%) with kidney renal clear cell carcinoma, and 55 samples of 280 patients (20%) with kidney renal papillary cell carcinoma. The NFI genes in kidney cancer were analyzed and depicted as oncoprints representing mutation, amplification, deep deletion, mRNA high,

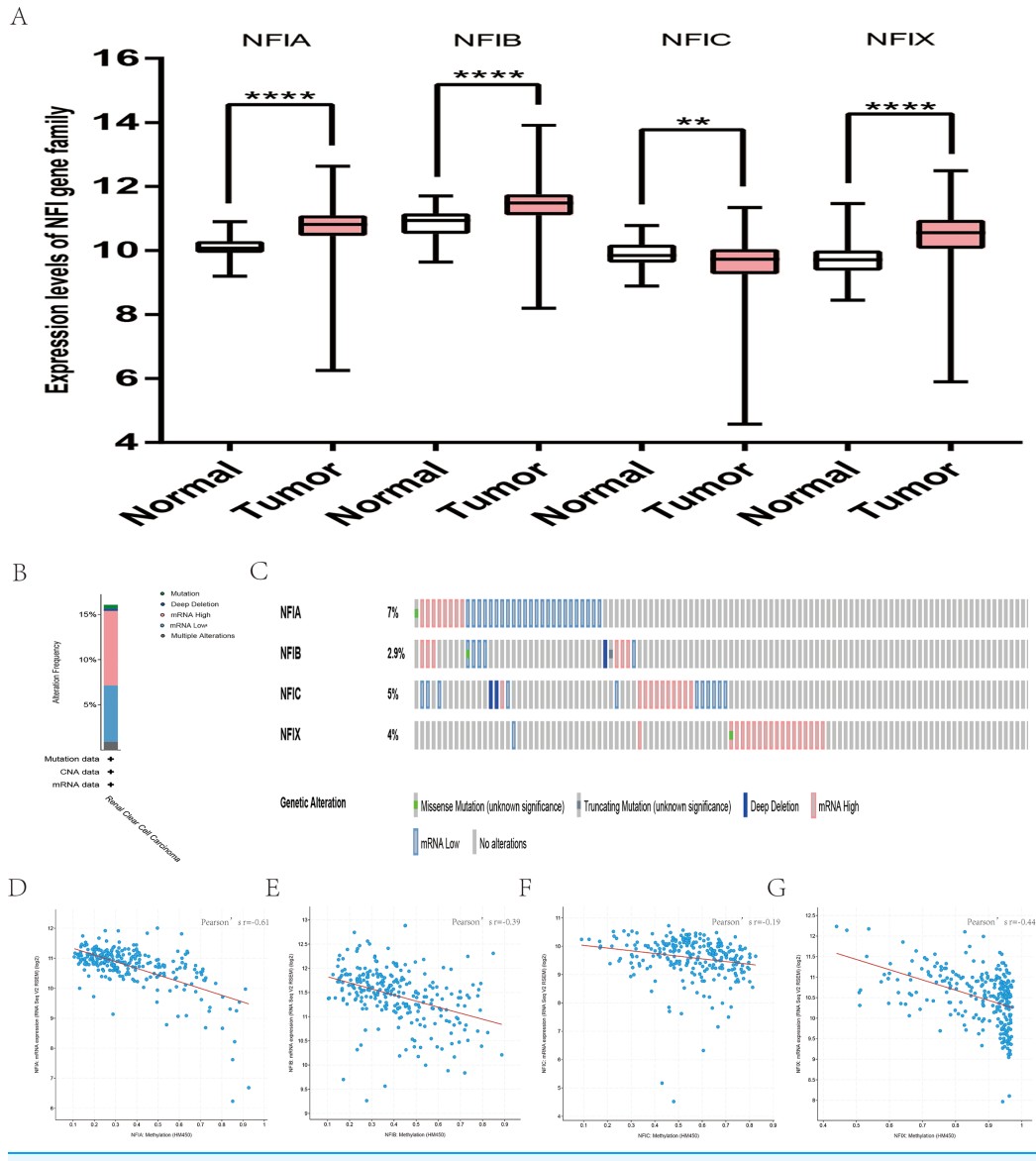

**Figure 8 Transcription levels, genomic analysis of the NFIs and regression analysis between the mRNA expression of the NFIs and its corresponding methylation in kidney renal clear cell carcinoma (A–G).** Box-whisker plots show the differences in transcript levels of the NFI family members between normal and tumors samples. The median value is represented by the middle line in the boxes. Statistical differences were examined by two-tailed student's *t*-test. **$p < 0.001$, ****$p < 0.0001$. NFI, Nuclear factor I; TCGA, The Cancer Genome Atlas.

mRNA low and multiple alterations (Figs. 7B, 7C, 8B, 8C, 9B and 9C). Survival analysis of the NFI genes with and without each gene alteration was conducted. Kidney renal clear cell carcinoma patients with NFIA gene alteration showed significantly poor OS and DFS compared with kidney renal clear cell carcinoma patients without NFIA gene alteration (Fig. S8). Kidney renal papillary cell carcinoma patients with NFIX gene alteration showed worse OS compared with kidney renal papillary cell carcinoma patients without NFIX gene alteration (Fig. S9). However, significant difference in survival for kidney chromophobe cell carcinoma patients was not observed (Fig. S7). In addition, the

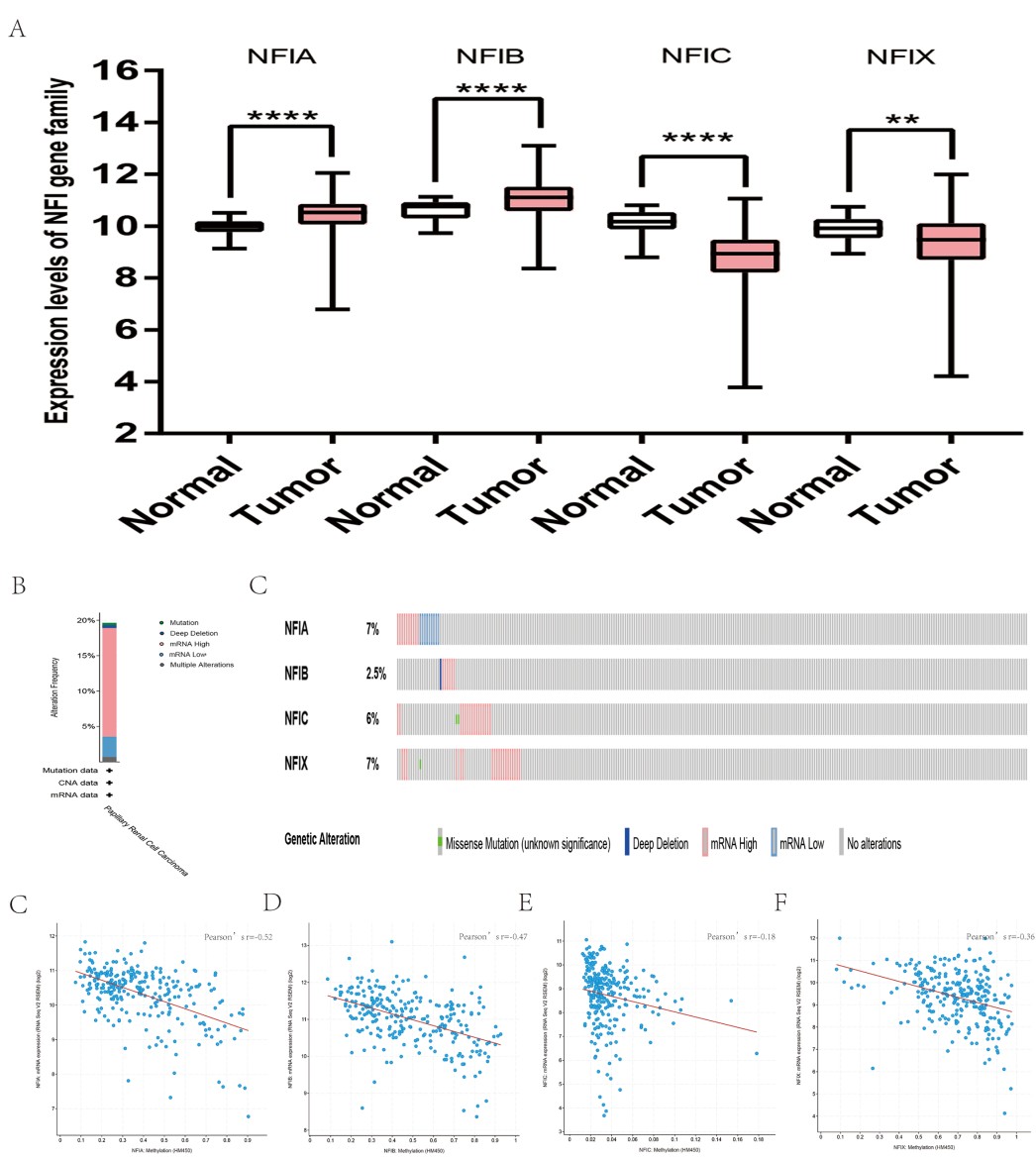

**Figure 9 Transcription levels, genomic analysis of the NFIs and regression analysis between the mRNA expression of the NFIs and its corresponding methylation in kidney renal papillary cell carcinoma (A–G).** Box-whisker plots show the differences in transcript levels of the NFI family members between normal and tumors samples. The median value is represented by the middle line in the boxes. Statistical differences were examined by two-tailed student's $t$-test. $^{**}p < 0.01$, $^{****}p < 0.0001$. NFI, Nuclear factor I; TCGA, The Cancer Genome Atlas.           

correlation between NFI gene expression and its DNA methylation was calculated using the cBioPortal online tool for kidney chromophobe cell carcinoma (TCGA, Firehose Legacy), kidney renal clear cell carcinoma (TCGA, Firehose Legacy), and kidney renal papillary cell carcinoma (TCGA, Firehose Legacy), and Pearson's correction was included. The results indicated significantly negative correlation between NFI gene expression and corresponding DNA methylation in kidney cancer. Regarding kidney chromophobe cell carcinoma (Figs. 7D–7G), regression analysis confirmed a moderately negative

**Table 12 Correlation of NFIs with survival outcomes in kidney cancer patients.**

| Gene | RNAseq ID | Survival outcome | No. of cases | Cut-off value | HR | 95% CI | p-Value |
|------|-----------|------------------|--------------|---------------|-----|--------|---------|
| (A) Correlation of NFIs with survival outcomes in kidney renal clear cell carcinoma patients | | | | | | | |
| NFIA | 4774 | OS | 530 | 1,758 | 0.5 | [0.37–0.68] | <0.001 |
|      |      | RFS | 117 | 1,865 | 3E+08 | [0–lnf.] | 0.017 |
| NFIB | 4781 | OS | 530 | 3,585 | 0.46 | [0.33–0.63] | <0.001 |
|      |      | RFS | 117 | 2,646 | 4.62 | [0.61–35.15] | 0.1 |
| NFIC | 4782 | OS | 530 | 3,627 | 0.73 | [0.54–0.98] | 0.036 |
|      |      | RFS | 117 | 3,868 | 2.47 | [0.69–8.89] | 0.15 |
| NFIX | 4784 | OS | 530 | 1,668 | 0.66 | [0.49–0.88] | 0.0054 |
|      |      | RFS | 117 | 1,711 | 10.49 | [1.38–80.04] | 0.0048 |
| (B) Correlation of NFIs with survival outcomes in kidney renal papillary cell carcinoma patients | | | | | | | |
| NFIA | 4774 | OS | 287 | 1,332 | 0.34 | [0.19–0.61] | <0.001 |
|      |      | RFS | 183 | 1,290 | 0.55 | [0.25–1.19] | 0.13 |
| NFIB | 4781 | OS | 287 | 2,122 | 0.29 | [0.16–0.55] | <0.001 |
|      |      | RFS | 183 | 2,149 | 0.36 | [0.17–0.77] | 0.006 |
| NFIC | 4782 | OS | 287 | 2,721 | 0.62 | [0.34–1.13] | 0.11 |
|      |      | RFS | 183 | 3,808 | 0.59 | [0.24–1.48] | 0.26 |
| NFIX | 4784 | OS | 287 | 1,268 | 2.28 | [1.26–4.13] | 0.0051 |
|      |      | RFS | 183 | 1,219 | 1.63 | [0.73–3.67] | 0.23 |

**Note:**
HR, hazard ratio; CI, confidence interval; OS, overall survival; RFS, relapse free survival.

correlation in NFIA (Pearson's $r = -0.4$), NFIC (Pearson's $r = -0.4$) and NFIX (Pearson's $r = -0.42$). Correlation between NFIB expression and its DNA methylation was not observed (Pearson's $r = -0.11$). In kidney renal clear cell carcinoma (Figs. 8D–8G), regression analysis indicated a strongly negative correlation in NFIA (Pearson's $r = -0.61$), a moderately negative correlation in NFIX (Pearson's $r = -0.44$), and a weakly negative correlation in NFIB (Pearson's $r = -0.39$). Correlation between NFIC expression and its methylation was not observed (Pearson's $r = -0.19$). In addition, regression analysis confirmed a moderately negative correlation in NFIA (Pearson's $r = -0.52$) and NFIB (Pearson's $r = -0.47$), a weakly negative correlation in NFIX (Pearson's $r = -0.36$), and no correlation in NFIC (Pearson's $r = -0.18$) for kidney renal papillary cell carcinoma (Figs. 9D–9G). Next, the prognostic significance associated with the expression of the NFI family members was evaluated using the KM plotter database (Nagy et al., 2018). The results showed low expression of the NFI family members predicted worse OS in kidney clear cell carcinoma. Reduced expression of NFIA and NFIX was significantly associated with better RFS. In kidney papillary cell carcinoma, decreased NFIB mRNA level was significantly correlated with worse OS and RFS. Decreased NFIA expression was also associated with worse OS but not RFS. Conversely, low NFIX expression predicted better OS. The details are shown in Tables 12A and 12B. However, the survival data for kidney chromophobe cell carcinoma were not available in the KM plotter or SurvExpress database.

## Transcription levels and prognostic significance of the NFI family members in other cancers

The Oncomine database showed significant differences in mRNA expression of NFIB and NFIC between cervical cancer and normal tissues (Fig. 1). The details are shown in Table S3. NFIB expression was downregulated in high grade cervical squamous and cervical squamous cell carcinoma analyzed according to Zhai and Scotto's datasets, respectively (*Scotto et al., 2008*; *Zhai et al., 2007*). In Biewenga's study (*Biewenga et al., 2008*), the NFIC mRNA expression was also reduced in cervical squamous cell carcinoma compared with normal tissues. However, the difference in expression of the NFI family members between tumors and normal tissues in the TCGA database could not be compared due to the lack of normal samples. Subsequently, the prognostic value associated with the expression of the NFI family members was determined using the KM plotter database. The results showed lower mRNA expression of NFIB, NFIC and NFIX predicted worse OS in cervical squamous cell carcinoma. In addition, elevated NFIA and NFIB mRNA levels were significantly associated with worse RFS (Table S4 File).

For colorectal cancer, analysis using the Oncomine database revealed significant difference only in NFIA and NFIB mRNA levels between tumor and normal samples (Table S5). Expression of NFIA and NFIB was downregulated in colorectal cancer according to the TCGA database. However, NFIA expression was increased in rectal mucinous adenocarcinoma and cecum adenocarcinoma in Kaiser's study (*Kaiser et al., 2007*). NFIB mRNA level was upregulated in colon adenoma according to Skrzypczak's dataset (*Skrzypczak et al., 2010*). Expression of NFIA, NFIB and NIFC was reduced in colorectal cancer tissues evaluated based on mRNA HiSeq expression data from the TCGA database (Fig. S10). Next, the prognostic significance associated with the expression of the NFI family members was evaluated using the KM plotter database. Only NFIX expression was associated with OS in colorectal cancer patients (Table S6).

For gastric cancer, NFIB and NFIX mRNA levels were not significantly different between tumor and normal tissues (Table S7). DErrico's dataset showed overexpression of the NFIB gene. However, NFIX mRNA level was significantly downregulated in gastric mixed adenocarcinoma (*D'Errico et al., 2009*). Due to the limited number of cases in the Oncomine database, 380 gastric cancers and 37 normal samples from the TCGA database were further used to confirm the potential expression differences of the NFI family members between tumors and normal tissues. NFIB expression was elevated in gastric cancer. However, the NFIC mRNA level was downregulated in gastric cancer compared with normal tissues (Fig. S11). Subsequently, to ascertain the prognostic value associated with the expression of the NFI family members in gastric cancer, OS, PPS and FP were evaluated using the KM plotter database. The prognostic effects of the four genes are shown in Table S8. Low expression of NFIC and NFIX predicted poor OS in gastric cancer patients. In addition, reduced NFIX expression was significantly associated with worse FP in gastric cancer patients. Next, the prognostic ability of NFI expression was investigated in different HER2 statuses of gastric cancer. As shown in Tables S8A–S8C, only reduced NFIX mRNA expression could predict worse OS, FP and PPS in the HER2-positive group.

However, in the HER2-negative group, NFIB and NFIC expression was significantly correlated with better OS and decreased NFIX expression was associated with better PPS.

For liver cancer, analysis using the Oncomine database revealed only NFIA mRNA level was significantly downregulated in tumor tissues according to Wurmbach's dataset (*Wurmbach et al., 2007*). Other genes of the NFI family did not show any significant difference between tumors and normal tissues (Table S9). NFIA, NFIB and NFIC expression levels were reduced in cancer tissues evaluated using mRNA HiSeq expression data from the TCGA database (Fig. S12). In the KM plotter database, high NFIA and NFIX expression predicted better OS and disease-specific survival (DSS) in liver cancer patients. Besides, decreased NFIC expression level was associated with poor PFS and NFIB expression was uncorrelated with OS in liver cancer patients (Table S10).

For prostate cancer, the Oncomine database was used to compare the mRNA expression levels of the NFI family members between cancer and normal tissues. The results showed NFIB expression was upregulated in prostate adenocarcinoma according to Wallace's dataset (*Wallace et al., 2008*), however, the opposite results were found in Tomlins's study (*Tomlins et al., 2007*). NFIC mRNA levels were downregulated in prostate carcinoma according to Luo2's dataset (*Luo et al., 2002*). NFIX expression was lower in prostate carcinoma in Varambally's study (*Varambally et al., 2005*), but higher in benign prostate hyperplasia according to Tomlins's dataset (*Tomlins et al., 2007*). The details are shown in Table S11. In the TCGA database, NFIA, NFIB and NFIC mRNA levels were significantly downregulated in prostate cancer patients. However, a significant upregulation of NFIX mRNA levels was observed in tumor tissues (Fig. S13). Alternatively, the prognostic significance associated with the expression of the NFI family members was determined using the SurvExpress database because survival data for prostate cancer was absent in the KM plotter database. The results showed no significant association between expression of the NFI family members and survival outcome in prostate cancer patients (Fig. S14).

## DISCUSSION

The NFI transcription factors play an important role in normal development and are related to human dysplasia. Due to the advancement of high-throughput sequencing technology, the NFI family has been found to play an important role in the development of various tumors. In this study, the mRNA expression levels of the NFI family members were comprehensively analyzed in various types of cancers using the Oncomine and TCGA databases. The results showed the mRNA expression levels of the NFI family were significantly downregulated in breast, bladder, lung, esophageal, and head and neck cancers compared with normal tissues. In addition, the transcriptional pattern of the NFI family was different among the three subtypes of kidney cancer. For example, the expression of NFIA and NFIB was reduced in kidney chromophobe cell carcinoma but not in papillary or clear cell carcinoma; however, mRNA expression levels were elevated. Furthermore, NFIB expression was increased in gastric cancer. Then, the cBioPortal online tool was used to investigate the underlying mechanism of dysregulated expression of the NFI family in breast, bladder, lung, esophageal, kidney, and head and neck cancers. Genomic analysis

showed the mRNA high percentage of NFIA, NFIB, NFIC and NFIX was higher than in other gene alterations, including gene mutation, amplification and deep deletion. Survival analysis indicated that almost none of the NFI genes with gene alterations were associated with OS or DFS. These findings indicate that NFI gene alterations might not independently influence its transcription in various tumors. In addition, by comparing the mRNA expression level of each NFI family member with its corresponding DNA methylation, a certain negative correlation was observed, indicating that methylation may be one reason for the decreased expression levels of the NFI family.

Reportedly, the NFI transcription factors are significantly associated with various clinicopathological features and survival outcomes in cancer patients. In bladder cancer, elevated NFIA mRNA expression was associated with T1 progressive bladder cancer compared with T1 nonprogressive tumors (Sharron Lin et al., 2014). Stringer et al. (2016) demonstrated that low NFIB mRNA expression was associated with increased astrocytoma grade and mesenchymal subtype of glioblastoma. Glioblastoma multiforme (GBM) patients with higher NFIB expression survived significantly longer than patients with lower NFIB expression. In another study, NFIX DNA hypermethylation was reportedly associated with significantly decreased NFIX expression and was related to shorter OS and RFS in patients with lung adenocarcinoma (Ge et al., 2018).

However, only a limited number of studies have investigated the prognostic significance of the NFI transcription factors in human cancers. In a previous study, high NFIA expression was shown an independent predictor of poor prognosis in esophageal squamous carcinoma, and high NFIB expression was a negative prognostic value in esophagogastric junction adenocarcinoma (Yang et al., 2018). Therefore, the mRNA expression of certain NFI transcription factors might correlate with survival outcomes in cancer patients. In the present study, high expression of NFIA, NFIB and NFIX was significantly associated with improved prognosis in breast cancer. In addition, these significant correlations were present specifically in the luminal A and HER2+ subtypes of breast cancer. Decreased NFIA mRNA expression indicated better OS, RFS and DMFS in breast cancer patients with HER2+ subtype but worse OS, RFS and DMFS in luminal A subtype patients. In lung cancer, expression of NFIA and NFIB was correlated with better prognosis. However, such correlations might only be applicable to lung adenocarcinomas but not squamous cell carcinoma. All four genes were significantly associated with poor prognosis in ovarian cancer. In gastric cancer, high NFIX expression was significantly correlated with better overall prognosis in gastric cancer and HER2+ gastric cancer, and marginally correlated with PPS in HER2-gastric cancer. Reduced mRNA expression of NFIA, NFIB and NFIX predicted better OS in bladder cancer. Furthermore, statistical correlations were found between mRNA expression of the different NFI family members and survival outcomes in head and neck, liver, kidney, cervical, esophageal and pancreatic cancers, as well as sarcoma. Taken together, the data indicate the NFI family members may be used as prognostic biomarkers in many cancers.

Recently, accumulating evidence indicated the NFI transcription factors have both oncogenic and tumor-suppressive potential, depending on the context. For example, NFIB, the most well studied NFI transcription factor, might be oncogenic in SCLC

(*Dooley et al., 2011*), melanoma, and breast cancer (*Fane et al., 2017*; *Liu et al., 2019*), but likely functions as a tumor suppressor in NSCLC (*Becker-Santos et al., 2016*), osteosarcoma (*Mirabello et al., 2015*), glioma and glioblastoma (*Stringer et al., 2016*; *Suzuki et al., 2015*). Similarly, NFIB and NFIC could function in an opposing role. The breast cancer cell line, MCF7, treated with NFIC siRNA, enhanced EMT, motility, migration and invasion (*Lee, Lee & Park, 2015*). Conversely, depletion of NFIB in p53-mutated triple-negative breast cancer cell lines MDA-MB-435, HCC1806 and BT-20, promoted cell death, cell cycle arrest, and enhanced sensitivity to docetaxel, a first-line chemotherapeutic drug used in breast cancer treatment (*Liu et al., 2019*). Furthermore, NFIA inhibited cell death and enhanced cell survival, proliferation, and migration in GBM by negatively regulating p53, p21 and PAI1 (*Kang et al., 2016*). In ESCC cells, NFIX overexpression inhibited cell proliferation, migration, and invasion (*Mao et al., 2015*). In the present study, based on the large databases in Oncomine and TCGA, the mRNA expression levels of the NFI family members were downregulated in various types of cancer. Genomic analysis showed the alterations in each NFI family gene were less frequent in various tumors and had little influence on survival outcomes. In addition, the correlation between NFI gene expression and its DNA methylation was calculated using the cBioPortal online tool for breast, bladder, lung, esophageal, kidney, and head and neck cancers. A certain negative correlation was observed, indicating that epigenetic alteration is an important mechanism of dysregulated NFI expression in human cancers. When generating Kaplan–Meier curves, statistical correlations were found between mRNA expression of the NFI family genes and survival outcomes in multiple tumors. Therefore, we hypothesized the NFI family might play an important role in tumor processes, and further investigation of the underlying molecular mechanisms in multiple cancers is necessary.

## CONCLUSION

In the present study, the mRNA expression levels, genetic alteration, DNA methylation, and prognostic significance of the NFI family were systematically analyzed in different human cancers using the Oncomine, TCGA, KM plotter, SurvExpress databases and cBioPortal for Cancer Genomics. Based on the large amount of data, the mRNA expression levels of the NFI family members were downregulated among various types of cancer and DNA hypermethylation may be an important cause of the downregulation. Furthermore, several of the NFI genes showed great prognostic significance for cancer patients. Therefore, additional studies are needed to further explore the detailed role of the NFI family in tumor initiation and development, which may confirm the NFI family members are promising therapeutic targets and novel prognostic biomarkers for human cancers.

### Funding

This work was supported by the National Natural Science Foundation of China (No. 81472806). The funders had no role in study design, data collection and analysis, decision to publish, or preparation of the manuscript.

## Grant Disclosures

The following grant information was disclosed by the authors:
National Natural Science Foundation of China: 81472806.

## Competing Interests

The authors declare that they have no competing interests.

## Author Contributions

- Yuexian Li conceived and designed the experiments, performed the experiments, analyzed the data, prepared figures and/or tables, authored or reviewed drafts of the paper, and approved the final draft.
- Cheng Sun analyzed the data, prepared figures and/or tables, authored or reviewed drafts of the paper, and approved the final draft.
- Yonggang Tan performed the experiments, authored or reviewed drafts of the paper, and approved the final draft.
- Lin Li analyzed the data, prepared figures and/or tables, and approved the final draft.
- Heying Zhang performed the experiments, prepared figures and/or tables, and approved the final draft.
- Yusi Liang performed the experiments, analyzed the data, authored or reviewed drafts of the paper, and approved the final draft.
- Juan Zeng analyzed the data, prepared figures and/or tables, and approved the final draft.
- Huawei Zou conceived and designed the experiments, authored or reviewed drafts of the paper, funding acquistion, and approved the final draft.

## Data Availability

The raw data are available in the Supplemental Files.

## Supplemental Information

Supplemental information for this article can be found online at http://dx.doi.org/10.7717/peerj.8816#supplemental-information.

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
