# Peer review of "Transcription levels and prognostic significance of the NFI family members in human cancers"

_PeerJ, doi:10.7717/peerj.8816_

## Round 0.1 · original submission · Major Revisions

Some concerns have to be addressed before accepting the manuscript since the Reviewers have quite different opinions.

Reviewer 1 ·

Basic reporting

There were no novel findings in the manuscript compared to the previous reports.

Experimental design

All results of this article is from Oncomine, TCGA, KM plotter and SurvExpress databases. There is no any further experimental evidence for the transcription level change in any cancer type. That could not help to verify the correlation of NFI family in human cancer.

Validity of the findings

The finding is without any novelty and lack of enough evidences to support this opinion.

Additional comments

Although database is a kind of strategy to find out the correlation of target gene and human cancer, the experimental data is more important to link the finding of database. Nuclear factor I transcription factors was already found the relationship of human cancer. The article did not support any more novel findings.

·

Basic reporting

Clear English is used throughout. The article is very well written.
Good literature references and sufficient field background/context are provided.
The authors used a professional article structure, figures and tables and the raw data is shared through supplemental tables and figures.
The article is self-contained with relevant results to the hypothesis that NFI transcription factors may be important in a variety if cancers.

Experimental design

The authors clearly defined the research question, which is relevant and meaningful. The knowledge gap being investigated has been well identified, and statements have been made as to how the study contributes to filling that gap. While only a review and mining of the current literature and databases, the authors make a good effort to point out the significance of their findings. The techniques used by the authors appear valid and consistently applied. They are described in good detail in the methods section.

Validity of the findings

The conclusions of the authors appear to be clearly consistent with the data presented. All of the data are from pre-existing databases and is freely available. The statistical methods used appear sound. The conclusions are clear though somewhat vague. No obvious speculation is present.

Reviewer 3 ·

Basic reporting

The manuscript is clear and english was easy to follow.
Literature references are scarce, at the introduccion, and can be improved but not a problem for a bioinformatic analysis.
The structure of the articles can be largely improved by proving deeper analyses.

Experimental design

The experimental design is poor since does not consider why the expression changes across human cancer or to which other variables are the expression correlated. This shows that the experimental design is quite superfluous.
Research question is well defined but not so relevant if not appropriately analyzed.
The investigation was not rigorous since statistical thinking is lacking. Suggestion are recommended in general comments.

Validity of the findings

Not novel (checking expression across cancers) and low impact if not complemented by deeper analyses.
Statistically, analyses does not sound.
Conclusion is meaningless if not deeper analysis is provided.

Additional comments

Review for the manuscript #38719 “Transcription levels and prognostic significance of the NFI family members in human cancers” received from PeerJ. In this work, the authors carry out a bioinformatic gene expression analysis of the family of 4 transcription factors in human cancers. For this, they used Oncomine, UCSC Xena, and SurvExpress. First, the authors check the differential expression of the genes across cancers to show that in about 1/5 of the analyses, these genes are altered. Then, they describe more details in some cancer types including
survival analyses related to the expression of these genes. The authors conclude that the NFI worth to be further studied. The manuscript is well written and easy to follow.

Although the analysis may have some value, I believe it lacks statistical rigor, deeper analysis, and predictions or plausible hypotheses. The following comments show details.

- The observations in figure 1 could be by random chance. If this is rando, why readers should care about NFI? So, what is the probability of observing differences as those shown in NFI gene family? The authors need to take several families of proteins and do the same analysis in order to show that the expected number of differences is higher in NFI than random chance.
- Survival analysis also needs to be done for other protein families in order to show whether survival is more affected by NFI expression than random proteins.
- The authors describe in text sections the difference in expression per cancer type. However, deeper analysis of why the expression seems to be altered is needed (otherwise is just a mere description of what is observed in Figure 1). Are the changes in expression related to copy number alterations? To mutations in specific genes? To subtypes within the same study? Correlated to other proteins? Associated to changes in methylation?
- Do the analysis encourage plausible biomedical hypotheses? This needs to be discussed.

---

## Round 0.2 · Minor Revisions

Reviewer 1 comments have not been properly considered. Consider reviewer 1 responses as the ones performed with the other 2 reviewers. The authors should address point by point those comments and should clearly describe the novelty of the study. Please, also check the revisions inserted in the manuscript, they need to be more accurate and revised since there are English errors.

---

## Round 0.3 · accepted · Accept

The manuscript can now be considered for publication.